# Stochastic Polyak Step-sizes and Momentum: Convergence Guarantees and Practical Performance

**Dimitris Oikonomou**
CS & MINDS
Johns Hopkins University
doikono1@jh.edu

**Nicolas Loizou**
AMS & MINDS
Johns Hopkins University
nloizou1@jh.edu

## Abstract

Stochastic gradient descent with momentum, also known as Stochastic Heavy Ball method (SHB), is one of the most popular algorithms for solving large-scale stochastic optimization problems in various machine learning tasks. In practical scenarios, tuning the step-size and momentum parameters of the method is a prohibitively expensive and time-consuming process. In this work, inspired by the recent advantages of stochastic Polyak step-size in the performance of stochastic gradient descent (SGD), we propose and explore new Polyak-type variants suitable for the update rule of the SHB method. In particular, using the Iterate Moving Average (IMA) viewpoint of SHB, we propose and analyze three novel step-size selections: MomSPS$_{\max}$, MomDecSPS, and MomAdaSPS. For MomSPS$_{\max}$, we provide convergence guarantees for SHB to a neighborhood of the solution for convex and smooth problems (without assuming interpolation). If interpolation is also satisfied, then using MomSPS$_{\max}$, SHB converges to the true solution at a fast rate matching the deterministic HB. The other two variants, MomDecSPS and MomAdaSPS, are the first adaptive step-size for SHB that guarantee convergence to the exact minimizer - without a priori knowledge of the problem parameters and without assuming interpolation. Our convergence analysis of SHB is tight and obtains the convergence guarantees of stochastic Polyak step-size for SGD as a special case. We supplement our analysis with experiments validating our theory and demonstrating the effectiveness and robustness of our algorithms.

## 1 Introduction

We consider the unconstrained finite-sum optimization problem,

$$\min_{x \in \mathbb{R}^d} \left[ f(x) = \frac{1}{n} \sum_{i=1}^{n} f_i(x) \right], \tag{1}$$

where each $f_i : \mathbb{R}^d \to \mathbb{R}$ is convex, smooth, and lower bounded by $\ell_i^*$. Let $X^*$ be the set of minimizers of (1). We assume that $X^* \neq \emptyset$ and we fix $x^* \in X^*$. This problem is the cornerstone of machine learning tasks, (Hastie et al., 2009), where $x$ corresponds to the model parameters, $f_i(x)$ represents the loss on the training point $i$, and the aim is to minimize the average loss $f(x)$ across training points.

When $n$ is large, stochastic gradient methods are the preferred methods for solving (1) mainly because of their cheap per iteration cost. One of the most popular stochastic algorithms for solving such large-scale machine learning optimization problems is stochastic gradient descent (SGD) with momentum, (Sutskever et al., 2013), a.k.a. stochastic heavy ball method (SHB) given by:

$$x^{t+1} = x^t - \gamma_t \nabla f_{S_t}(x^t) + \beta_t(x^t - x^{t-1}). \tag{SHB}$$

where $S_t \subseteq [n]$ a random subset of data-points (mini-batch) with cardinality $B$ sampled independently at each iteration $t$, and $\nabla f_{S_t}(x^t) = \frac{1}{B} \sum_{i \in S_t} \nabla f_i(x^t)$ is the mini-batch gradient. Here $\gamma_t > 0$ is the step-size/learning rate at iteration $t$ while $\beta_t \geq 0$ represents the momentum parameter.

When the momentum parameter $\beta_t = 0, \forall t \geq 0$, then the update rule SHB is equivalent to the well-studied mini-batch SGD, $x^{t+1} = x^t - \gamma_t \nabla f_{S_t}(x^t)$, (Robbins and Monro, 1951), which has been efficiently analyzed under different properties of problem (1) and different step-size selections $\gamma_t$ (Nemirovski and Yudin, 1983; Nemirovski et al., 2009; Hardt et al., 2016; Needell et al., 2016; Nguyen et al., 2018; Gower et al., 2019; 2021). Additionally, when the cardinality of the random subset $S_t$ is $B = n$, then the update rule of SHB is equivalent to the deterministic heavy ball method (HB) proposed by Polyak (1964), as a way to improve the convergence behavior of deterministic Gradient Descent (GD).

There is a rich literature on the convergence of SHB and HB in different scenarios. In Polyak (1964), it was proved that for a specific choice of the step-size $\gamma$ and the momentum parameter $\beta$, the HB method enjoys an accelerated linear convergence when minimizing strongly convex quadratic functions while more recently, Ghadimi et al. (2015) proved a global sublinear convergence guarantee for HB for convex and smooth functions. In the stochastic setting, several works focus on convergence guarantees of SHB under constant step-size and momentum parameters (Ma and Yarats, 2019; Kidambi et al., 2018; Yan et al., 2018; Gitman et al., 2019; Liu et al., 2020). However, in practical scenarios, these choices suffer from a prohibitively expensive and time-consuming hyper-parameter tuning process. This has motivated a large body of research on the development of adaptive SHB - a method that adapts their parameters using information collected during the iterative process. Such analysis is challenging, and the current adaptive versions of SHB either focus on the full batch setting (deterministic) (Barré et al., 2020; Saab Jr et al., 2022; Wang et al., 2023), or assume that an interpolation condition is satisfied (Schaipp et al., 2023b) or focus on moving averaged gradient (a different form of momentum) (Wang et al., 2023).

Previous studies in the fully stochastic (non-interpolated) scenario have predominantly concentrated on analyzing adaptive versions of SGD, with limited attention given to developing adaptive variants for the SHB. In this work, we take inspiration from the recently introduced and highly efficient Polyak-type adaptive step-sizes for SGD and investigate its applicability and extension to SHB.

## 1.1 MAIN CONTRIBUTIONS

Our main contributions are summarized below.

$\diamond$ **Efficient Polyak Step-sizes via IMA viewpoint.** We explain and illustrate by experiment (see Figure 1) why naively using $\text{SPS}_{\max}$ of Loizou et al. (2021) as a step-size $\gamma_t$ in the update rule of SHB is not robust, leading to divergence even in simple problems. To resolve this issue, we provide an alternative way of selecting Polyak-type step-sizes for SHB via the Iterate Moving Average viewpoint from Sebbouh et al. (2021). Through our approach, we propose three novel adaptive step-size selections, namely $\text{MomSPS}_{\max}$, MomDecSPS, MomAdaSPS. Each of the proposed step-sizes depends on the choice of the momentum parameter $\beta$ adding further stability to SHB and comes with specific benefits over their constant step-size counterparts or other adaptive variants of SHB.

$\diamond$ **$\text{MomSPS}_{\max}$: Convergence of SHB in non-interpolated setting.** Our first step-size selection of SHB is $\text{MomSPS}_{\max}$, which has a similar structure to $\text{SPS}_{\max}$ of Loizou et al. (2021) but includes also $(1 - \beta)$ in its expression. For this choice, we provide convergence guarantees for SHB to a neighborhood of the solution for convex and smooth problems. Our analysis provides the first convergence guarantees of adaptive SHB using Polyak-type step-size. Previous works on Polyak step-size with momentum in the stochastic setting have guarantees either only under the interpolation setting (Schaipp et al., 2023b) or for a moving averaged gradient momentum (Wang et al., 2023). In addition, as a corollary of our main theoretical results, we show that $\text{MomSPS}_{\max}$ under the interpolation setting and in deterministic scenarios (full batch) converges to the true solution at a fast rate matching the deterministic HB.

$\diamond$ **Convergence of SHB to exact solution via MomDecSPS and MomAdaSPS.** Inspired by two recent Polyak-type step-size selections for SGD, the DecSPS in Orvieto et al. (2022) and AdaSPS in Jiang and Stich (2023), we propose two new ways for tuning the step-size for SHB. These are MomDecSPS and MomAdaSPS. Our analysis provides the first $O(1/\sqrt{T})$ convergence guarantees to the exact solution in the non-interpolated regime for a Polyak-type adaptive variant of SHB. Our proposed update rules converge for any choice of momentum parameter $\beta \in [0, 1)$, which makes them particularly useful in practical scenarios.

| Step-size | Assumptions / Setting | Adaptive | Exact Convergence | Rate |
|-----------|----------------------|----------|-------------------|------|
| Constant (Liu et al., 2020) | Knowledge of $L$ | ✗ | ✗ | $O(\frac{1}{T} + \hat{\sigma}^2)$ |
| IMA (Sebbouh et al., 2021) | Knowledge of $L$ | ✗ | ✓ | $O(\frac{1}{T} + \hat{\sigma}^2)$ |
| ALR-SMAG (Wang et al., 2023) | Moving Averaged Gradient | ✓ | ✗ | $O(\frac{1}{T} + \sigma^2)$ |
| MomSPS$_{\max}$ (Thm 3.2) | Restriction on $\beta$ | ✓ | ✗ | $O(\frac{1}{T} + \sigma^2)$ |
| MomDecSPS (Thm 3.6) | Bounded Iterates | ✓ | ✓ | $O(\frac{1}{\sqrt{T}})$ |
| MomAdaSPS (Thm 3.7) | Bounded Iterates | ✓ | ✓ | $O(\frac{1}{T} + \frac{\sigma}{\sqrt{T}})$ |

Table 1: Summary of the considered step-sizes and the corresponding theoretical results in the stochastic setting. All the rates are given for convex and smooth functions. The quantity of convergence in our rates is $\mathbb{E}[f(\overline{x}^T) - f(x^*)]$, where $\overline{x}^T = \frac{1}{T}\sum_{t=0}^{T} x^t$. Here $\hat{\sigma}^2 = \mathbb{E}\|\nabla f_i(x^*)\|^2 < \infty$ and $\sigma^2 = \mathbb{E}[f_i(x^*) - \ell_i^*] < \infty$. The "Exact Convergence" column refers to convergence to the exact solution $x^*$ with no interpolation assumption.

◇ **Tight Convergence Guarantees.** All of our convergence guarantees are *true* generalizations of the theoretical analysis of SGD using SPS$_{\max}$, DecSPS, and AdaSPS. That is, if $\beta = 0$ (no momentum) in the update rule of SHB, our theorems obtain as a special case the best-known convergence rates of Polyak-type step-size for SGD, highlighting the tightness of our analysis. See also Table 1 for a summary of our main complexity results and a comparison with closely related works.

◇ **Further Convergence Results.** As a byproduct of our theoretical analysis, we provide two interesting corollaries: a *novel analysis of constant step-size SHB* (as a corollary of our Theorem on MomSPS$_{\max}$) and the first *robust convergence of SHB* via our theorem on MomAdaSPS. For a constant step-size, the Corollary 3.5 of our Theorem 3.2 allows larger step-sizes than the analysis of SHB in Liu et al. (2020) and provides convergence without assuming the restrictive bounded variance condition (there exist $q > 0$ such that $\mathbb{E}\|\nabla f_i(x) - \nabla f(x)\|^2 < q$). In addition, via Theorem 3.7 we provide the first robust convergence of adaptive SHB that guarantees convergence to the exact solution and automatically adapts to whether our problem is interpolated or not. That is, if interpolation is assumed, then the rate of SHB with MomAdaSPS is the same as the rate of SHB with MomSPS$_{\max}$ (or constant step-size), and if no interpolation is assumed, then it matches the rate of SHB with MomDecSPS. The analysis achieves the best-known rates in both settings.

◇ **Numerical Evaluation.** In Section 4, we verify our theoretical results via numerical experiments on various problems, demonstrating the effectiveness and practicality of our approach. An open-source implementation of our method is available at `https://github.com/dimitris-oik/MomSPS`.

## 2 EXPLORING THE INTERPLAY OF SPS AND HEAVY BALL MOMENTUM

In this section, we present the expression of Stochastic Polyak Step-size (SPS) and its different variants. We illustrate that a naive combination of SPS with momentum is not robust, leading to divergence in simple problems. We explain how we resolve this issue using the Iterate Moving Average viewpoint of SHB and propose three adaptive Polyak-type step-sizes for SHB.

### 2.1 BACKGROUND ON STOCHASTIC POLYAK STEP-SIZE

The deterministic Polyak step-size (PS) was first introduced by Polyak (1987) as an efficient step-size selection for GD for solving convex optimization problems. It has the following expression: $\gamma_t = \frac{f(x^t) - f(x^*)}{\|\nabla f(x^t)\|^2}$ which is obtained by minimizing an upper bound of the quantity $\|x^{t+1} - x^*\|^2$ in the analysis of GD. Since its original proposal, the PS has been successfully used in the analysis of deterministic subgradient methods in different settings with favorable convergence guarantees (Boyd et al., 2003; Davis et al., 2018; Hazan and Kakade, 2019). PS requires the prior knowledge of $f(x^*)$, which might look like a strong assumption. However, as shown in Boyd et al. (2003), this is known in several applications, including finding a point in the intersection of convex sets and positive semi-definite matrix completion.

Inspired by the convergence of PS in the deterministic setting, Loizou et al. (2021) has effectively modified the Polyak step-size for the stochastic setting, achieving convergence rates comparable to those of finetuned SGD. The proposed stochastic Polyak step-size (SPS) has several benefits,

including independence on parameters of the problem (e.g., $L$-smoothness or $\mu$ strong convexity) and competitive performance in over-parametrized models. More specifically, Loizou et al. (2021) proposed the $\text{SPS}_{\max}$ given below[1]:

$$\gamma_t = \min\left\{\frac{f_{S_t}(x^t) - \ell^*_{S_t}}{c\|\nabla f_{S_t}(x^t)\|^2}, \gamma_b\right\}. \tag{$\text{SPS}_{\max}$}$$

Here, $\gamma_b > 0$ is a bound that restricts SPS from being very large and is essential to ensure convergence to a small neighborhood around the solution and $c > 0$ is a positive constant that depends on the function class the objective $f$ belongs to.

As mentioned in Orvieto et al. (2022), the $\text{SPS}_{\max}$ comes with strong convergence guarantees and competitive performance; however, it has one main drawback when used in non-over-parameterized regimes: It can guarantee convergence only to a neighborhood of the solution. For this reason, Orvieto et al. (2022) suggests a decreasing variant of the original $\text{SPS}_{\max}$ named DecSPS, given by:

$$\gamma_t = \frac{1}{c_t}\min\left\{\frac{f_{S_t}(x^t) - \ell^*_{S_t}}{\|\nabla f_{S_t}(x^t)\|^2}, c_{t-1}\gamma_{t-1}\right\}, \tag{DecSPS}$$

where $c_t$ is an increasing sequence of positive real numbers and $c_{-1} := c_0$ and $\gamma_{-1} = \gamma_b > 0$. The authors proved that SGD with DecSPS and $c_t = \sqrt{t+1}$, converges with a sublinear rate $O(1/\sqrt{T})$ for convex and smooth functions with bounded iterates (i.e., $D^2 := \max_{t\in[T-1]}\|x^t - x^*\|^2 < \infty$).

More recently, in Jiang and Stich (2023), another decreasing variant of SPS was introduced, named AdaSPS:

$$\gamma_t = \min\left\{\frac{f_{S_t}(x^t) - \ell^*_{S_t}}{c\|\nabla f_{S_t}(x^t)\|^2}\frac{1}{\sqrt{\sum_{s=0}^t f_{S_s}(x^s) - \ell^*_{S_s}}}, \gamma_{t-1}\right\}, \tag{AdaSPS}$$

where $\gamma_{-1} = +\infty$ and $c > 0$. For convex and $L$-smooth functions with bounded iterates, it can be shown that SGD with AdaSPS converges to an exact solution with a rate $O\left(\frac{\tau^2}{T} + \frac{\tau\sigma}{\sqrt{T}}\right)$, where $\tau = 2cLD^2 + \frac{1}{c}$. The interesting aspect of the convergence of AdaSPS is that it provides a robust result for SGD, meaning that the method recovers the best bounds for both the interpolated ($\sigma = 0$) and non-interpolated regimes.

## 2.2 NAIVE SPS IN SHB

All of the above variants of SPS were proposed and analyzed for SGD (SHB with no momentum). With the increased popularity of momentum in machine learning, one can naturally ask the following question: *Is it possible to combine SPS with momentum?* This question was partially answered in Wang et al. (2023) when they proposed a Polyak-type step-size for a moving averaged gradient (MAG) momentum but not for the SHB. Their MAG framework is not equivalent to SHB when their stepsizes are adaptive and while their proposed stepsize reduces to the original $\text{SPS}_{\max}$ stepsize when $\beta = 0$, their guarantees do not reduce to the original $\text{SPS}_{\max}$ guarantees for convex functions.

The most straightforward approach is to directly apply the $\text{SPS}_{\max}$ in the SHB update rule. Let us call this update rule $SPS_{\max}$ *with naive momentum.* Unfortunately, this approach does not necessarily lead to convergence for natural choices of momentum parameter $\beta \in (0, 1)$, as shown in Figure 1.[2] In this experiment, even for simple convex and smooth problems like a logistic regression with synthetic data, the naive rule fails to converge when $\beta$ gets larger (a typical choice for momentum parameter is $\beta = 0.9$). This indicates that a more careful step-size selection is needed, which may also depend

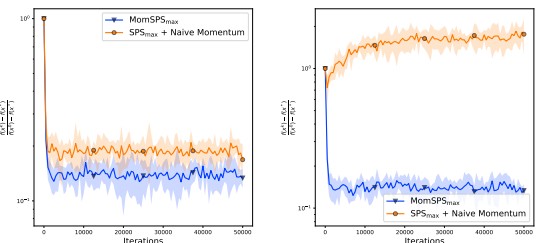

Figure 1: Comparison of $\text{MomSPS}_{\max}$ versus $\text{SPS}_{\max}$ with naive momentum for different momentum parameters $\beta$ on a logistic regression problem. Left: $\beta = 0.2$, Right: $\beta = 0.5$

---

[1]Originally Loizou et al. (2021) use $f^*_{S_t}$ instead of the lower bound $\ell^*_{S_t}$. The lower bound is due to Orvieto et al. (2022), which proves that the more relaxed lower bound can still lead to the same convergence guarantees.

[2]See also Figure 9 in Appendix for more values of $\beta$.

on the momentum parameter. In Figure 1, we compare the SPSmax with naive momentum with one of our proposed and analyzed step-size selection $\text{MomSPS}_{\text{max}}$. As seen in Figure 1, when $\beta$ is small, then both $\text{SPS}_{\text{max}}$ with naive momentum and SHB with $\text{MomSPS}_{\text{max}}$ have similar behavior. When $\beta = 0$, the two methods have identical performance as both are reduced to the same method: SGD with $\text{SPS}_{\text{max}}$. However, as $\beta$ gets larger, the performance of $\text{SPS}_{\text{max}}$ with naive momentum gets worse and less stable, and at some point, it diverges. On the other hand, in this example, SHB with $\text{MomSPS}_{\text{max}}$ converges for any of the selected $\beta$.

## 2.3 Iterative Moving Average: Balance between SPS and Momentum

Having explained how naively combining $\text{SPS}_{\text{max}}$ with heavy ball momentum can lead to divergence, in this section, we leverage the iterate moving-average (IMA) viewpoint of SHB from Sebbouh et al. (2021) to propose SPS-type adaptations that depend on momentum parameter as well for SHB. As we will see later, this viewpoint provides stability and robustness to the proposed update rules.

**Iterate Moving Average (IMA).** Sebbouh et al. (2021) provide the IMA as an alternative way of expressing the update rule of SHB and explain how the IMA formulation is crucial in comparing SHB and SGD as it allows to establish connections between the step-sizes of the two methods. The Iterate Moving Average method is given by the following update rule:

$$z^{t+1} = z^t - \eta_t \nabla f_{S_t}(x^t), \quad x^{t+1} = \frac{\lambda_{t+1}}{\lambda_{t+1}+1} x^t + \frac{1}{\lambda_{t+1}+1} z^{t+1},$$

where $z^0 = x^0$, $\eta_t > 0$ and $\lambda_t \geq 0$. As proved in Sebbouh et al. (2021), if for any $t \in \mathbb{N}$, holds $1 + \lambda_{t+1} = \frac{\lambda_t}{\beta_t}$ and $\eta_t = (1 + \lambda_{t+1})\gamma_t$ then the $x_t$ iterates of the IMA method are equal to the $x_t$ iterates produced by the SHB method. For completeness, we include the proof of this statement in Appendix C.3.

In our proposed methods, we select the most common setting of constant momentum ($\beta_t = \beta$). Considering the equivalence between IMA and SHB under the correct parameter selection, let us present the following proposition that will allow us to obtain convergence guarantees for SHB via a convergence analysis of IMA.

**Proposition 2.1.** *If $\lambda_t = \lambda \geq 0$ then assuming that $\beta = \frac{\lambda}{1+\lambda}$ and $\gamma_t = (1 - \beta)\eta_t$ the $x_t$ iterates of the IMA method are equal to the $x_t$ iterates produced by the SHB method.*

Using the above proposition, let us provide the following corollary that explains the derivation of our proposed step-size selections for SHB.

**Corollary 2.2.** *Let the step-size $\eta_t$ in IMA be one of the previously proposed Polyak step-sizes: $\text{SPS}_{\text{max}}$, DecSPS or AdaSPS and let $\lambda_t = \lambda$. Then via Proposition 2.1, the SHB has constant momentum parameter $\beta_t = \beta$ and the step-sizes are $\text{MomSPS}_{\text{max}}$, MomDecSPS or MomAdaSPS respectively.*

**Three New Adaptive Step-sizes for SHB.** Using the IMA viewpoint and Proposition 2.1 and Corollary 2.2 let us present three Polyak-type step-size for SHB.

**(i) $\text{MomSPS}_{\text{max}}$.** Let us start with the variant associated with the $\text{SPS}_{\text{max}}$. That is, via Corollary 2.2 if we use $\text{SPS}_{\text{max}}$ as a step-size for IMA, then this is equivalent to using the following step-size in the update rule of SHB:

$$\gamma_t = (1 - \beta) \min \left\{ \frac{f_{S_t}(x^t) - \ell_{S_t}^*}{c\|\nabla f_{S_t}(x^t)\|^2}, \gamma_b \right\}. \tag{$\text{MomSPS}_{\text{max}}$}$$

Here, the parameter $\gamma_b > 0$ has the same purpose as in the original $\text{SPS}_{\text{max}}$, and it is a bound that restricts $\text{MomSPS}_{\text{max}}$ from being very large and is essential to ensure convergence to a neighborhood around the solution. It is clear from its expression that $\text{MomSPS}_{\text{max}}$ has the same form as $\text{SPS}_{\text{max}}$ but multiplied by a *correcting momentum factor* $1 - \beta$. This follows from Corollary 2.2. In practice, this small change allows the SHB to be more "stable" for different momentum parameters, as we show in Figure 1. Using $\text{MomSPS}_{\text{max}}$ and a restriction on the momentum parameter $\beta$ in Section 3, we establish $O(1/T)$ convergence for SHB up to a neighborhood of the solution for convex and smooth functions.

Our following two proposed step-size selections are variants of the two decreasing variants of Polyak step-sizes DecSPS and AdaSPS. As such, we can prove SHB's convergence to the exact solution instead of a neighborhood. More importantly, using MomDecSPS or MomAdaSPS, we can provide convergence guarantees for any choice of the momentum coefficients $\beta \in [0, 1)$ making SHB fully adaptive (no tuning necessary).

**(ii) MomDecSPS.** Firstly, we propose an adaptation of DecSPS with momentum. We call this step-size selection MomDecSPS, and it is given by:

$$\gamma_t = \min\left\{ \frac{(1-\beta)[f_{S_t}(x^t) - \ell^*_{S_t}]}{c_t\|\nabla f_{S_t}(x^t)\|^2}, \frac{\gamma_{t-1}c_{t-1}}{c_t} \right\}, \qquad \text{(MomDecSPS)}$$

where $\gamma_{-1} := \gamma_b > 0$ is a step-size bound and $c_t = c\sqrt{t+1}$ for $t \geq 0$ with $c_{-1} := c_0 = c > 0$ be a constant to regulate the step-size. Note that $\gamma_t$ is indeed decreasing. This step-size is adaptive and, in its definition, does not require knowledge of any function properties (e.g., smoothness/strong convexity constants). In Section 3 we prove that SHB with MomDecSPS and constant momentum $\beta \in [0, 1)$ converges to exact solution with a rate $O(1/\sqrt{T})$.

**(iii) MomAdaSPS.** Similar to the previous two steps, let us propose the following adaptation of AdaSPS with momentum. We call this MomAdaSPS, and is given by:

$$\gamma_t = \min\left\{ \frac{(1-\beta)[f_{S_t}(x^t) - \ell^*_{S_t}]}{c\|\nabla f_{S_t}(x^t)\|^2 \sqrt{\sum_{s=0}^{t} f_{S_s}(x^s) - \ell^*_{S_s}}}, \gamma_{t-1} \right\}. \qquad \text{(MomAdaSPS)}$$

By definition, this is a decreasing step-size ($\gamma_t \leq \gamma_{t-1}$). In Section 3, we prove that SHB with MomAdaSPS also converges to the exact solution for any choice of $\beta \in [0, 1)$. More specifically, we show that MomAdaSPS converges to the exact solution with a rate $O\left(\frac{1}{T} + \frac{\sigma}{\sqrt{T}}\right)$. Thus, when we are in the interpolation regime (i.e., $\sigma^2 = 0$), the convergence of SHB with MomAdaSPS matches SHB with MomSPS$_{max}$ while when the setting does not satisfy interpolation condition, this became equivalent to the $O(1/\sqrt{T})$ of MomDecSPS. Following the terminology of Jiang and Stich (2023), our results are the first *robust* adaptive step-size selection for SHB, as it can automatically adapt to the optimization setting (interpolation vs. non-interpolation).

Let us close this section by mentioning two remarks related to the above three step-size selections:

**Remark 2.3.** The "correcting factor" $1 - \beta$ is outside the minimum in the MomSPS$_{max}$ while for the decreasing variants MomDecSPS and MomAdaSPS, it only appears in the first term of the minimum. This follows from Proposition 2.1 and the derivation of Corollary 2.2 and is explained in detail in Appendix C.3. In Appendix F.3, via experiments, we also test other variants of the above step-sizes with the correcting factor outside of the minimum. We observe that the other choices do not have good practical performance.

**Remark 2.4.** For no momentum, i.e., $\beta = 0$, all three proposed step-size choices, MomSPS$_{max}$, MomDecSPS and MomAdaSPS are reduced to SPS$_{max}$, DecSPS and AdaSPS and our convergence analysis recovers the convergence guarantees for SGD showing the tightness of our approach.

## 3 CONVERGENCE ANALYSIS

In this section, we present the convergence results for SHB with all of the proposed Polyak step-sizes. For the formal definitions and helpful lemmas, see Appendix C. The proofs can be found in Appendix D. For all of our results, we make the following assumption:

**Assumption 3.1** (Finite optimal objective difference)**.**

$$\sigma^2 := \mathbb{E}_{S_t}[f_{S_t}(x^*) - \ell^*_{S_t}] = f(x^*) - \mathbb{E}_{S_t}[\ell^*_{S_t}] < \infty$$

This assumption was first introduced in Loizou et al. (2021) and Orvieto et al. (2022). Note that $\sigma^2 < \infty$ when all $f_i$ are lower bounded like we have assumed. We say that problem (1) is *interpolated* if $\sigma^2 = 0$. If the interpolation condition is satisfied then there exists $x^* \in X^*$ such

that $f(x^*) = f_{S_t}(x^*) = \ell_{S_t}^*$ for all $S_t \subseteq [n]$. Many modern machine learning models satisfy this condition. Examples include non-parametric regression (Liang and Rakhlin, 2020) and over-parameterized deep neural networks (Zhang et al., 2021; Ma et al., 2018).

## 3.1 CONVERGENCE TO A NEIGHBORHOOD OF THE SOLUTION

We start with the analysis of SHB with MomSPS$_{\max}$.

**Theorem 3.2.** *Assume that each $f_i$ is convex and $L_i$-smooth. Then, the iterates of SHB with MomSPS$_{\max}$ with $c = 1$ and $\beta \in \left[0, \frac{\alpha}{2\gamma_b - \alpha}\right)$ where $\alpha = \min\left\{\frac{1}{2L_{\max}}, \gamma_b\right\}$ and $L_{\max} = \max_i\{L_i\}$, converge as*

$$\mathbb{E}[f(\overline{x}^T) - f(x^*)] \leq \frac{C_1\|x^0 - x^*\|^2}{T} + C_2\sigma^2,$$

*where $\overline{x}^T = \frac{1}{T}\sum_{t=0}^{T-1} x^t$ and the constants $C_1 = \frac{1-\beta}{\alpha\beta + \alpha - 2\beta\gamma_b}$ and $C_2 = \frac{2\gamma_b - \alpha\beta - \alpha}{\alpha\beta + \alpha - 2\beta\gamma_b}$.*

Firstly, let us note that both constants $C_1$ and $C_2$ in Theorem 3.2 are positive since the denominator of both is positive because $\beta \in [0, 1)$ and the numerator of $C_2$ is positive because $\gamma_b \geq \alpha > 0$. The above result shows that MomSPS$_{\max}$ has a sublinear convergence to a neighborhood matching the best-known rate for SHB in the convex setting, see Liu et al. (2020). Moreover, when $\beta = 0$, the update rule of SHB with MomSPS$_{\max}$ becomes equivalent to SGD with SPS$_{\max}$ (no momentum) and the result of Theorem 3.2 reduces to convergence guarantees provided in Loizou et al. (2021), only with a slightly better neighborhood of convergence (here when $\beta = 0$ we have $C_2 = (2\gamma_b - \alpha)/\alpha$ while in the original it is $C_2 = 2\gamma_b/\alpha$).

In addition, note that in Theorem 3.2, there is a restriction in the momentum coefficient. This stems from the technique used in the proof (which forces $C_1$ and $C_2$ to be positive). However, this restriction is not vital as in practical scenarios, the method converges even when $\beta$ is selected outside the given interval. Moreover, let us highlight that the constants $C_1$ and $C_2$ are increasing when viewed as functions of $\beta$ in the given interval (see also Figure 14). This means that our theory suggests that $\beta = 0$ (no momentum) is the best theoretical choice, which is typical in many works on stochastic methods with momentum (Wang et al., 2023; Loizou and Richtárik, 2020). This is not ideal but does not undermine the importance of Theorem 3.2, as this is the first result showing the convergence of SHB with SPS in a non-interpolated setting.

From Theorem 3.2, we can also deduce rates for the deterministic and interpolated regimes.

**Corollary 3.3** (Deterministic Heavy Ball Method). *Assume that $f$ is convex and $L$-smooth. Then HB with $\beta \in [0, \alpha/(2\gamma_b - \alpha))$ and*

$$\gamma_t = (1 - \beta) \min\left\{\frac{f(x^t) - f(x^*)}{\|\nabla f(x^t)\|^2}, \gamma_b\right\}, \qquad \text{(MomPS}_{\max})$$

*where $\alpha = \min\left\{\frac{1}{2L}, \gamma_b\right\}$, converges as $\min_{t\in[T]}\{f(x^t) - f(x^*)\} \leq \frac{(1-\beta)\|x^0 - x^*\|^2}{(\alpha\beta + \alpha - 2\beta\gamma_b)T}$.*

Note that in the setting of Corollary 3.3, we have used $f(x^*)$ instead of a lower bound $\ell^*$. This is in accordance with previous work in the deterministic setting (Polyak, 1987). We highlight that in Hazan and Kakade (2019) (PS) and Wang et al. (2023) (ALR-MAG), there is no upper bound in the step-size for the deterministic setting. This is due to the fact that in these works, the step-size is derived as a minimizer of an upper bound of the quantity $\|x^t - x^*\|^2$. Our step-size MomPS$_{\max}$ is *not* a minimizer of such bound when $\beta > 0$. Nevertheless, when $\beta = 0$ (no momentum), by setting $\gamma_b = \infty$ we recover the corresponding result in Hazan and Kakade (2019). In our setting, we cannot set $\gamma_b = \infty$ when $\beta > 0$ because the term $\beta\gamma_b$ appears in the denominator of our convergence result.

**Corollary 3.4** (Interpolation). *Assume interpolation ($\sigma^2 = 0$) and let all assumptions of Theorem 3.2 be satisfied. Then SHB with MomSPS$_{\max}$ and $\beta \in [0, \alpha/(2\gamma_b - \alpha))$ where $\alpha = \min\left\{\frac{1}{2L_{\max}}, \gamma_b\right\}$ converges as $\mathbb{E}[f(\overline{x}^T) - f(x^*)] \leq \frac{(1-\beta)\|x^0 - x^*\|^2}{(\alpha\beta + \alpha - 2\beta\gamma_b)T}$, where $\overline{x}^T = \frac{1}{T}\sum_{t=0}^{T-1} x^t$.*

In the closely related works Loizou et al. (2021) ($\text{SPS}_{\max}$) and Wang et al. (2023) (ALR-SMAG), there is no upper bound in the proposed step-sizes for the interpolated setting. In our analysis, we focus on the fully stochastic setting (interpolation is only a corollary), and the result of Corollary 3.4 shows a saddle connection between the momentum parameter and the parameter $\gamma_b$ (we cannot select $\gamma_b = \infty$ when momentum is used). Nevertheless, via our corollary, when $\beta = 0$ by setting $\gamma_b = \infty$, we can recover the corresponding result in Loizou et al. (2021).

Another Corollary of Theorem 3.2 is a novel analysis for SHB with constant step-size.

**Corollary 3.5** (Constant Step-size). *Let all assumptions of Theorem 3.2 be satisfied. If $\gamma_b \leq \frac{1}{2L_{\max}}$, then SHB with $\gamma \leq \frac{1-\beta}{2L_{\max}}$ and $\beta \in [0,1)$ converges as $\mathbb{E}[f(\overline{x}^T) - f(x^*)] \leq \frac{\|x^0 - x^*\|^2}{T\gamma} + \sigma^2$, where $\overline{x}^T = \frac{1}{T}\sum_{t=0}^{T-1} x^t$.*

Note that in Corollary 3.5, there is no restriction that depends on the smoothness parameter in the momentum parameter $\beta$. We allow to have $\beta \in [0,1)$. A similar result is obtained in Liu et al. (2020), where the authors establish the convergence rate $O\left(\frac{f(x^0)-f(x^*)}{T} + \frac{L\gamma\hat{\sigma}^2}{1-\beta}\right)$ for the quantity $\frac{1}{T}\sum_{t=0}^{T-1} \mathbb{E}[\|\nabla f_{S_t}(x^t)\|^2]$ when $\gamma \leq \frac{(1-\beta)^2}{L}\min\left\{\frac{1}{4-\beta+\beta^2}, \frac{1}{2\sqrt{2\beta+2\beta^2}}\right\}$. Here $\hat{\sigma}^2$ is assumed to be a bound of the variance $\mathbb{E}_t \|\nabla f_{S_t}(x^t) - \nabla f(x^t)\|^2 \leq \hat{\sigma}^2$. In comparison, in our analysis, we provide the same asymptotic rate $O(1/T)$ for the common in the convex setting quantity $\mathbb{E}[f(\overline{x}^T) - f(x^*)]$. By comparing the two results, our step-size is larger when $\beta \geq \sqrt{5} - 2 \approx 0.236$. However, in our result, the neighborhood of convergence is constant and does not depend on the step-size $\gamma$ or the momentum coefficient. For a numerical comparison of SHB with the two different constant step-sizes, see Appendix F.2.

## 3.2 CONVERGENCE TO THE EXACT SOLUTION

In this section, we provide theoretical results for the adaptive decreasing step-sizes, MomDecSPS and MomAdaSPS. The main advantage of these decreasing step-sizes is that we can guarantee convergence to the exact solution while keeping the main adaptiveness properties. Due to the nature of the decreasing step-sizes, there is no restriction in the momentum parameter, as was the case with Theorem 3.2. For the results of this section, we make the extra assumption of *bounded iterates*, i.e., we assume that $D^2 = \max_{t \in [T]} \|x^t - x^*\|^2$ is finite. This is a standard assumption for several adaptive step-sizes, see: (Reddi et al., 2018; Ward et al., 2020; Orvieto et al., 2022; Jiang and Stich, 2023).

**Theorem 3.6.** *Assume that each $f_i$ is convex and $L_i$-smooth. Let $\overline{x}^T = \frac{1}{T}\sum_{t=0}^{T-1} x^t$, $\alpha = \min\left\{\frac{1}{2L}, \gamma_b\right\}$ and $D^2 = \max_{t \in [T]} \|x^t - x^*\|^2$. Then, the iterates of SHB with MomDecSPS with $c = 1$ and $\beta \in [0,1)$ converge as:*

$$\mathbb{E}[f(\overline{x}^T) - f(x^*)] \leq \frac{2\beta[f(x^0) - f(x^*)]}{(1-\beta)T} + \frac{(1+\beta)D^2}{(1-\beta)\alpha\sqrt{T}} + \frac{2\sigma^2}{\sqrt{T}}.$$

Notice that when there is no momentum, i.e., for $\beta = 0$, Theorem 3.6 recovers the original result of SGD with DecSPS from Orvieto et al. (2022). To our knowledge, the only other result that guarantees convergence to the exact solution for SHB in the non-interpolated regime appears in Sebbouh et al. (2021). In particular, Sebbouh et al. (2021) proves that for $\gamma_t = \frac{\eta_t}{1+\lambda_{t+1}}$ and $\beta_t = \frac{\lambda_t}{1+\lambda_t}$ where $\eta_t = \frac{\eta}{\sqrt{t+1}}$ with $0 \leq \eta \leq \frac{1}{4L_{\max}}$ and $\lambda_t = \frac{1}{2}\sum_{i=0}^{t-1}\sqrt{\frac{t+1}{i+1}}$ with $\lambda_0 = 0$, SHB converges as $\mathbb{E}[f(x^t) - f(x^*)] \leq O(\log t/\sqrt{t})$. This result only holds for a specific choice of $\beta_t$, it needs knowledge of $L_{\max}$, and only guarantees $O(\log t/\sqrt{t})$ rate. However, it does not make a bounded iteration assumption, and it shows convergence of the last iterate. In contrast, our convergence of SHB with MomDecSPS holds for any choice of $\beta \in [0,1)$, the step-size is adaptive and does not depend on $L_{\max}$, and it achieves a rate of $O(1/\sqrt{k})$.

Next, we present the theoretical guarantees for MomAdaSPS.

**Theorem 3.7.** *Assume that each $f_i$ is convex and $L_i$-smooth. Then, the iterates of SHB with MomAdaSPS and $\beta \in [0, 1)$ converge as*

$$\mathbb{E}[f(\overline{x}^T) - f(x^*)] \leq \frac{\tau^2}{T} + \frac{\tau\sigma}{\sqrt{T}},$$

*where $\overline{x}^T = \frac{1}{T}\sum_{t=0}^{T-1} x^t$, $\tau = \frac{\beta\sqrt{f(x^0) - f(x^*)}}{1-\beta} + \frac{(1+\beta)cLD^2}{2(1-\beta)} + \frac{1}{2c}$ and $D^2 = \max_{t\in[T]} \|x^t - x^*\|$.*

As in our previous theorems, when there is no momentum ($\beta = 0$), Theorem 3.7 recovers the convergence guarantees of AdaSPS from Jiang and Stich (2023). Let us highlight that our convergence guarantees of SHB with MomAdaSPS offers the first *robust* step-size selection for SHB, in the sense that it can automatically adapt to the optimization setting. More specifically, for no interpolation, our result has a rate of $O(1/\sqrt{T})$, which matches the best-known rate for SHB while if we assume interpolation ($\sigma^2 = 0$), then we are able to achieve a rate of $O(1/T)$ which matches the best-known rate for SHB in the convex setting (see Liu et al. (2020) and Theorem 3.2). Furthermore, under interpolation, Theorem 3.7 improves the convergence of Corollary 3.4. In particular, when $\sigma^2 = 0$, Theorem 3.7 reaches the rate $O(1/T)$ for *any* $\beta \in [0, 1)$, while in Corollary 3.4 this is possible only under tighter restrictions on $\beta$.

## 4 NUMERICAL EXPERIMENTS

In this section, we test our proposed algorithms in deterministic and stochastic convex problems as well as in training popular deep neural networks (DNNs). Our experiments are designed to highlight the benefits of our momentum variants over the vanilla (no-momentum) SPS.

**Deterministic Setting.** For the deterministic setting, we focus on the least squares problem. The loss function in this case is given by $f(x) = \frac{1}{2}\|Ax - b\|^2$, where $A \in \mathbb{R}^{n \times d}$ and $b \in \mathbb{R}^n$. In our experiments, we follow the setting of Wang et al. (2023) and choose $n = d = 1000$ while the matrix $A$ has been generated according to Lenard and Minkoff (1984) such that the condition number of $A^T A$ is $10^4$. We test the following algorithms: Gradient Descent (GD), Nesterov's Accelerated Gradient Descent (AGD) (Nesterov, 2018), Gradient Descent with Polyak's step-size

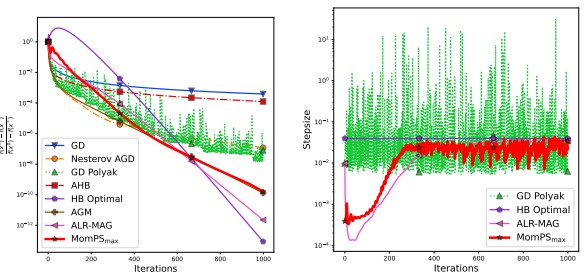

Figure 2: Comparison of deterministic algorithms for least squares. Left: Relative error, Right: Step-sizes.

(Polyak, 1987), Adaptive Heavy Ball (AHB) (Saab Jr et al., 2022), Polyak's Heavy Ball (HB) (Polyak, 1964), Accelerated Gradient Method (AGM) (Barré et al., 2020), ALR-MAG (Wang et al., 2023) and finally our step-size from Corollary 3.3, called (MomPS$_{max}$), which guarantees $O(1/T)$ rate. For the least squares problem, there is an established theory from Polyak (1964) that guarantees acceleration of HB with optimal step-size and momentum coefficient given by $\beta^* = (\sqrt{L} - \sqrt{\mu})^2/(\sqrt{L} + \sqrt{\mu})^2$ and $\gamma^* = (1 + \sqrt{\beta^*})/L$. For ALR-MAG and SHB with MomPS$_{max}$, we have used the optimal momentum $\beta^*$ for direct comparison with optimal HB. Moreover, for GD Polyak, ALR-MAG, and MomPS$_{max}$, we have used the precomputed $f(x^*)$ (via GD), and for MomPS$_{max}$, we have chosen $\gamma_b = 100$ (for other choices, see Appendix F.4). The results in Figure 2 show that MomPS$_{max}$ has performance comparable to AGM and ALR-MAG which are faster than vanilla Polyak step-size after the initial iterations. Furthermore, to explore the behavior of the various adaptive step-sizes, we plot the Polyak, the ALR-SMAG, the MomPS$_{max}$, and the optimal HB step-sizes. We can see that the original Polyak step-size oscillates around the optimal HB step-size while MomPS$_{max}$ converges from below to the optimal.

**Stochastic Setting: Convex Problems.** Here, we compare our step-sizes with previous works in the stochastic setting. As noticed initially for SPS$_{max}$ (Loizou et al., 2021), the value of the upper bound $\gamma_b$ that results in good convergence depends on the problem and requires careful parameter

tuning. To alleviate this problem, Loizou et al. (2021) uses a smoothing procedure that prevents large fluctuations in the step-size across iterations. That is, $\gamma_b^t = \tau^{b/n}\gamma_{t-1}$ for each iteration $t$ where $\tau = 2$, $b$ is the batch-size, and $n$ is the number of examples. We use the same smoothing trick for $\gamma_b$ in the implementation of the proposed methods. Similar smoothing procedures have been used in Tan et al. (2016); Vaswani et al. (2019b).

We considered multi-class logistic regression applied to commonly used benchmark datasets from the LIBSVM repository (Chang and Lin, 2011). We separate our experiments between two classes of step-sizes: non-decreasing step-sizes (which might guarantee convergence to a neighborhood of the solution) and decreasing step-sizes (which guarantee convergence to the exact solution). For the first class of step-sizes we test the following algorithms: SGD (Nesterov, 2018), SHB with the constant step-size and $\beta = 0.9$, ADAM as described in Kingma and Ba (2015), the warm-up version of ALR-SMAG with the hyper-parameters suggested in Wang et al. (2023), the SPS$_{max}$, the SPS$_{max}$ with naive momentum as described in Section 2.2 and our proposed MomSPS$_{max}$. For the Polyak-based step-sizes such as ALR-SMAG, SPS$_{max}$, and MomSPS$_{max}$, we select $\ell_i^* = 0$ and $c = 1$. For SHB, ALR-SMAG, MomSPS$_{max}$, SPS$_{max}$, with naive momentum, we use momentum $\beta = 0.9$. All the convex experiments are run for 100 epochs and for 5 trials. We plot the average of the trials and the standard deviation. Since there are no standard train/test splits, and due to the small sizes of the datasets, we present training loss and accuracy curves only. We present the comparison of these methods in Figure 3. In all experiments, we observe that the performance of our proposed MomSPS$_{max}$ significantly outperforms the other methods in both training loss and test accuracy. For the decreasing variants, we test AdaGrad-Norm (Duchi et al., 2011), SGD with DecSPS, SGD with AdaSPS, and our proposed SHB with MomDecSPS and SHB with MomAdaSPS. We present the outcome of this comparison in Figure 4. MomDecSPS and MomAdaSPS outperform their no-momentum counterparts in both training loss and test accuracy.

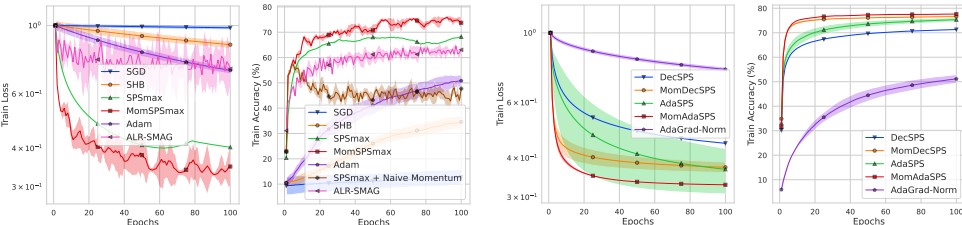

Figure 3: LibSVM dataset: vowel, Batch size: 52

Figure 4: LibSVM dataset: letter, Batch size: 1500

**MomSPS for DNNs.** In our final experiment, we go beyond the convex setting of our convergence guarantees, and we test MomSPS$_{max}$ on the training of DNNs. We consider non-convex minimization for multi-class classification using deep network models on the CIFAR 10 and CIFAR 100 datasets (Krizhevsky et al., 2009). We use two standard image-classification architectures: ResNet18 and ResNet34, (He et al., 2016). For space concerns, we report only the ResNet34 experiments in the main paper and relegate the ResNet18 to the Appendix. In these experiments we run MomSPS$_{max}$ with $c = 0.4$. We present the results for CIFAR 10 in Figure 5 and for CIFAR 100 in Figure 6. We observe that MomSPS$_{max}$ consistently outperforms its no-momentum counterpart SPS$_{max}$ and has competitive generalization performance compared to other popular optimizers.

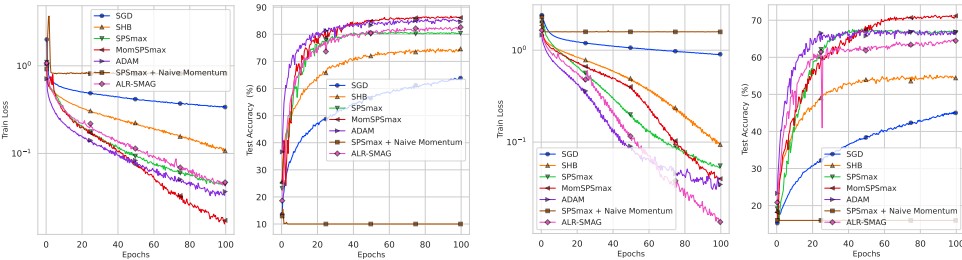

Figure 5: Resnet 34 on CIFAR 10

Figure 6: ResNet 34 on CIFAR 100

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

# Supplementary Material

The Supplementary Material is organized as follows: In Appendix A, we have more details on related work on adaptive and momentum methods. In Appendix B, we include the main pseudo-codes for our algorithms. In Appendix C, we give the basic definitions and lemmas as well as the basic theory of IMA. Appendix D presents the proofs of the theoretical guarantees from the main paper. In Appendix F, we describe in detail our experimental setup and provide additional experiments.

## CONTENTS

## A    FURTHER RELATED WORK

**Adaptive methods.**    Two of the first adaptive algorithms are AdaGrad (Duchi et al., 2011) and RMSProp (Hinton et al., 2012). The very popular algorithm Adam (Kingma and Ba, 2015) was introduced as a momentum extension of RMSProp, and its more recent weight-decay variant AdamW (Loshchilov and Hutter, 2019) is the de facto optimizer for deep neural networks. Investigating the convergence guarantees of adaptive methods across various settings continues to be an active area of research (Vaswani et al., 2020; Ward et al., 2020; Li and Orabona, 2019; Shi et al., 2022; Défossez et al., 2022; Choudhury et al., 2024b; Defazio and Jelassi, 2022).

More recently, a new line of work for adaptive algorithms has appeared, inspired by Polyak step-sizes. Some attempts for efficiently generalizing the Polyak step-size from the deterministic setting to the stochastic were made in Rolinek and Martius (2018); Prazeres and Oberman (2021); Berrada et al. (2020). Loizou et al. (2021) was the first work proposing $\mathrm{SPS}_{\max}$ and providing strong convergence guarantees in different settings, including strongly convex, convex, and non-convex functions. Further extensions with decreasing variants were proposed in Orvieto et al. (2022); Jiang and Stich (2023). Many recent works propose SPS-type update rules for solving optimization problems in different settings. For example, extensions of SPS to proximal setting analyzed in Schaipp et al. (2023a), SPS variants for mirror descent presented in D'Orazio et al. (2023), and SPS for federated learning proposed in Mukherjee et al. (2023). There are also strong connections between SPS and Model-Based approaches (Asi and Duchi, 2019a;b; Chadha et al., 2022). For further results related to SPS, see Gower et al. (2022); Li et al. (2023); Garrigos et al. (2023) and Abdukhakimov et al. (2023).

**Momentum with constant step-size.**    Polyak (1964) introduced the concept of momentum for the Gradient Descent (GD) method. He showed that for strongly convex quadratic problems, the momentum method provably accelerates GD with optimal momentum coefficient $\beta^* = (\sqrt{L} - \sqrt{\mu})^2/(\sqrt{L} + \sqrt{\mu})^2$ and $\gamma^* = (1 + \sqrt{\beta^*})^2/L$ where $\mu$ is the strong convexity constant and $L$ the smoothness constant. Ghadimi et al. (2015) proved a global sublinear convergence guarantee for HB for convex and smooth functions and global linear convergence when the function is also smooth and strongly convex. Moreover, Nesterov (1983) introduced Nesterov's Accelerated Gradient Descent (AGD) method, where he shows acceleration over GD for strongly convex and smooth as well as convex and smooth objectives. Note that both of these methods require knowledge of both $\mu$ and $L$. In the stochastic setting, Liu et al. (2020) provides new convergence guarantees for SHB with constant step-size. In particular, it shows that for both strongly convex and non-convex objectives, SHB enjoys the same convergence bound as SGD. Sebbouh et al. (2021), among other things, shows that in the smooth and convex setting SHB converges in expectation at the last iterate with a rate of $O(1/T)$ to a neighborhood of the minimum and at a $O(\log T/\sqrt{T})$ rate to the minimum exactly. For these guarantees, one needs knowledge of the problem parameters (such as the smoothness constant), and their results hold for a specific selection of the momentum parameter. Momentum has also been successfully applied in linear systems, see Loizou and Richtárik (2020) where it provides convergence guarantees of SGD, stochastic Newton, stochastic proximal point, and stochastic dual subspace ascent with momentum for consistent linear systems.

**Adaptive Momentum.**    In this paragraph, we review the bibliography for adaptive methods with momentum. Momentum can be either SHB or Nesterov's acceleration, and in this case, the adaptivity can be either the step-size or the momentum coefficient. In the deterministic setting, Barré et al. (2020) proposed the Accelerated Gradient Method (AGM), an adaptive algorithm built upon AGD that guarantees acceleration over GD. AGM approximates the strong convexity constant $\mu$ by the inverse of the classical Polyak Step-size. However, their algorithm still requires the knowledge of $L$ (but not $\mu$). Furthermore, Saab Jr et al. (2022) suggests an Adaptive Heavy Ball (AHB) algorithm where it approximates the constants $\mu$ and $L$ iteratively in each step based on the formula of the optimal constants of HB, but it does not show any acceleration over HB or GD. In the stochastic setting, Wang et al. (2023) propose an adaptive algorithm for a moving averaged gradient momentum, named ALR-SMAG. It shows that ALR-SMAG enjoys a linear convergence rate for semi-strongly convex and smooth functions. Schaipp et al. (2023b) proposes a new adaptive learning rate that can be combined with any momentum-based method. Finally, Zeng et al. (2023) proposes adaptive variants for SHB in linear systems solvers.

**On Technical Assumptions for Convergence.** In the literature of stochastic optimization problems, there are several assumptions on the noise of the stochastic estimators that typically are made on top of smoothness and convexity to prove the convergence of stochastic optimization algorithms.

For example, many works (Recht et al., 2011; Rakhlin et al., 2012; Shamir and Zhang, 2013; Nguyen et al., 2018) assume bounded gradients, i.e., that is, there exists a $M \in \mathbb{R}$ such that $\mathbb{E} \|\nabla f_i(x)\|^2 \leq M$. While this might look like a natural assumption, in the unconstrained setting, it contradicts the assumption of strong convexity leading to convergence guarantees that hold for an empty set of problems (Nguyen et al., 2018; Gower et al., 2019; 2021). A much more relaxed assumption used in the literature is the growth condition on the stochastic gradients, (Bertsekas and Tsitsiklis, 1996; Schmidt and Roux, 2013; Vaswani et al., 2019a). It states that there exist constants $\rho \in \mathbb{R}$ and $\delta \in \mathbb{R}$ such that $\mathbb{E} \|\nabla f_i(x)\|^2 \leq \rho \|\nabla f(x)\|^2 + \delta$. Based on a strong growth condition ($\delta = 0$), Schmidt and Roux (2013) were the first to establish linear convergence of SGD, with Vaswani et al. (2019a) later showing that SGD can find a first-order stationary point as efficiently as full gradient descent in non-convex settings. Similar conditions have also been proved and used in the analysis of decentralized variants of SGD (Koloskova et al., 2020; Assran et al., 2019). More recently, a line of work that uses smoothness (via expected smoothness/residual conditions) to provide closed-form expressions for the values of $\rho$ and $\delta$ of growth condition was able to provide tight convergence analysis of several stochastic algorithms, including SGD (Gower et al., 2019; 2021), variance reduced methods (Khaled et al., 2023), stochastic algorithms for min-max optimization (Loizou et al., 2021; 2020; Gorbunov et al., 2022; Choudhury et al., 2024a).

In the convex regime, the original analysis of SGD with SPS of Loizou et al. (2021) was one of the first papers that did not require any additional assumptions to guarantee convergence for SGD. Following the convergence guarantees of Loizou et al. (2021), we highlight that our proposed analysis of SHB with Polyak step-size does not require any additional assumptions for guaranteeing convergence. To the best of our knowledge, as mentioned in the main paper, our approach provides the first analysis of SHB without the restrictive bounded variance and growth conditions.

For proving convergence of SHB, Liu et al. (2020) makes the strong assumption of bounded variance and assumes knowledge of the smoothness $L$ for tuning the parameters. Another work on theoretical guarantees for SHB is Sebbouh et al. (2021), where a last iterate convergence is proved for a specific choice of $\gamma_t$ and $\beta_t$, but also requires knowledge of $L$ and only achieves rate $O(\log T/\sqrt{T})$ for the decreasing step-size variant. Furthermore, the authors of Wang et al. (2022) propose a Polyak-based stepsize for a moving averaged gradient momentum variant, different from SHB, and provide guarantees for the deterministic and stochastic setting for strongly convex objectives. Finally, Schaipp et al. (2023b) assumes a stochastic convex and *interpolated* regime for their results.

# B  PSEUDO-CODES

In this section, we include the pseudo-codes of the SHB with the proposed stochastic Polyak step-sizes for a better understanding of the methods and easier comparison with other algorithms.

**SHB with MomSPSmax.**  We start with the pseudo-code for SHB with $\text{MomSPS}_{\max}$. In Theorem 3.2 we established convergence to a neighborhood for $\beta \in \left[0, \frac{\alpha}{2\gamma_b - \alpha}\right)$ where $\alpha = \min\left\{\frac{1}{2L_{\max}}, \gamma_b\right\}$ and for $c = 1$. However, in practice, it seems to have convergence even for different choices of $c$ or even if $\beta$ is chosen outside of this bound, so we provide the pseudo-code for a general $\beta \in [0,1)$ and $c$. Thus, the user must provide the momentum coefficient $\beta$, the upper bound $\gamma_b$, the constant $c$, and the lower bounds $\ell_{S_t}^*$. In practice, the usual choices are $\beta = 0.9$, $c = 1$, and $\ell_{S_t}^* = 0$.

---

**Algorithm 1** SHB with $\text{MomSPS}_{\max}$

---

1: **Parameters:** $\beta \in [0,1)$, $\gamma_b > 0$, $c > 0$, $\ell_{S_t}^*$ lower bounds
2: **Initialization:** $x^0 \in \mathbb{R}^d$, $x^{-1} = x^0$
3: **for** $t = 0, 1, 2, \ldots$ **do**
4:     Choose uniformly at random $S_t \subseteq \{1, \ldots, n\}$
5:     $\gamma_t = (1 - \beta) \min\left\{\frac{f_{S_t}(x^t) - \ell_{S_t}^*}{c\|\nabla f_{S_t}(x^t)\|^2}, \gamma_b\right\}$
6:     $x^{t+1} = x^t - \gamma_t \nabla f_{S_t}(x^t) + \beta(x^t - x^{t-1})$
7: **end for**

---

**SHB with MomDecSPS.**  For the decreasing step-sizes, Theorems 3.6 and 3.7 make no assumption on $\beta \in [0,1)$. For SHB with MomDecSPS, the user needs to provide the momentum coefficient $\beta$, the upper bound $\gamma_b$, the constant $c$ and the lower bounds $\ell_{S_t}$, like the previous algorithm. In practice, the usual choices are $c = 1$ and $\ell_{S_t}^* = 0$. Note that when $t = 0$, the step-size is equal to

$$\gamma_0 = \min\left\{\frac{(1 - \beta)[f_{S_0}(x^0) - \ell_{S_0}^*]}{c\|\nabla f_{S_0}(x^0)\|^2}, \frac{\gamma_b c}{c}\right\} = (1 - \beta) \min\left\{\frac{f_{S_0}(x^0) - \ell_{S_0}^*}{c\|\nabla f_{S_0}(x^0)\|^2}, \gamma_b\right\},$$

which is equal to $\text{MomSPS}_{\max}$ for $t = 0$.

---

**Algorithm 2** SHB with MomDecSPS

---

1: **Parameters:** $\beta \in [0,1)$, $\gamma_{-1} = \gamma_b > 0$, $c > 0$, $c_{-1} = c > 0$, $c_t = c\sqrt{t + 1}$ for $t \geq 0$, $\ell_{S_t}^*$ lower bounds
2: **Initialization:** $x^0 \in \mathbb{R}^d$, $x^{-1} = x^0$
3: **for** $t = 0, 1, 2, \ldots$ **do**
4:     Choose uniformly at random $S_t \subseteq \{1, \ldots, n\}$
5:     $\gamma_t = \min\left\{\frac{(1 - \beta)[f_{S_t}(x^t) - \ell_{S_t}^*]}{c_t\|\nabla f_{S_t}(x^t)\|^2}, \frac{\gamma_{t-1} c_{t-1}}{c_t}\right\}$
6:     $x^{t+1} = x^t - \gamma_t \nabla f_{S_t}(x^t) + \beta(x^t - x^{t-1})$
7: **end for**

---

**SHB with MomAdaSPS.**  Finally, for SHB with MomAdaSPS the requirements are less since the user only needs to provide the momentum coefficient $\beta$, the constant $c$ and the lower bounds $\ell_{S_t}^*$. Furthermore, according to Jiang and Stich (2023) one can set $c = \frac{1}{\sqrt{f_{S_0}(x^0) - \ell_{S_0}^*}}$ after randomly choosing $S_0 \subseteq [n]$ in the first iteration. Moreover, the minimum with respect to $\gamma_{t-1}$ is to ensure that the step-size is decreasing.

---

**Algorithm 3** SHB with MomAdaSPS

---

1: **Parameters:** $\beta \in [0, 1)$, $\gamma_{-1} = +\infty$, $c > 0$, $\ell^*_{S_t}$ lower bounds
2: **Initialization:** $x^0 \in \mathbb{R}^d$, $x^{-1} = x^0$
3: **for** $t = 0, 1, 2, \ldots$ **do**
4: $\quad$ Choose uniformly at random $S_t \subseteq \{1, \ldots, n\}$
5: $\quad \gamma_t = \min \left\{ \dfrac{(1-\beta)[f_{S_t}(x^t) - \ell^*_{S_t}]}{c\|\nabla f_{S_t}(x^t)\|^2 \sqrt{\sum_{s=0}^{t} f_{S_s}(x^s) - \ell^*_{S_s}}}, \gamma_{t-1} \right\}$
6: $\quad x^{t+1} = x^t - \gamma_t \nabla f_{S_t}(x^t) + \beta(x^t - x^{t-1})$
7: **end for**

---

Note that for all the above algorithms, one needs only lower bounds $\ell^*_{S_t}$, which can be chosen equal to $\ell^*_{S_t} = 0$ in practice. Of course, if the true infima $f^*_{S_t} = \inf_{x \in \mathbb{R}^d} f_{S_t}$ are known, then one should use them because it will lead to better convergence in terms of the neighborhood. Furthermore, notice that we start by initializing $x^{-1} = x^0$, where $x^0$ is the given starting point. This is because the first step of SHB (i.e. for $t = 0$) is just an iteration of SGD because we have no previous iterate $x^{-1}$, that is $x^1 = x^0 - \gamma_0 \nabla f_{S_0}(x^0) + \beta(x^0 - x^{-1}) = x^0 - \gamma_0 \nabla f_{S_0}(x^0)$.

# C    TECHNICAL PRELIMINARIES

## C.1    BASIC DEFINITIONS

In this section, we present some basic definitions we use throughout the paper.

**Definition C.1** (Convexity). A differentiable function $f : \mathbb{R}^d \to \mathbb{R}$ is convex if

$$f(x) \geq f(y) + \langle \nabla f(y), x - y \rangle,$$

for all $x, y \in \mathbb{R}^d$.

**Definition C.2** ($L$-smooth). A differentiable function $f : \mathbb{R}^d \to \mathbb{R}$ is $L$-smooth if there exists a constant $L > 0$ such that

$$\|\nabla f(x) - \nabla f(y)\| \leq L \|x - y\|,$$

or equivalently

$$|f(x) - f(y) - \langle \nabla f(y), x - y \rangle| \leq \frac{L}{2} \|x - y\|^2$$

for all $x, y \in \mathbb{R}^d$.

## C.2    BASIC LEMMAS

Here we have gathered useful lemmas for various Polyak related step-sizes. These lemmas are used repeatedly in the proofs of the main results.

**Lemma C.3** ((Loizou et al., 2021)). *Suppose each $f_i$ is $L_i$-smooth, then the step-size of $SPS_{\max}$ satisfies:*

$$\alpha = \min\left\{\frac{1}{2cL_{\max}}, \gamma_b\right\} \leq \gamma_t \leq \gamma_b \text{ and } \gamma_t^2 \|\nabla f_{S_t}(x^t)\|^2 \leq \gamma_t [f_{S_t}(x^t) - \ell_{S_t}^*], \quad (2)$$

*where $L_{\max} = \max_{i \in [n]}\{L_i\}$.*

**Lemma C.4** (Lemma 1, (Orvieto et al., 2022)). *Suppose each $f_i$ is $L_i$-smooth, then the step-size of DecSPS satisfies*

$$\min\left\{\frac{1}{2c_t L_{\max}}, \frac{c_0 \gamma_b}{c_t}\right\} \leq \gamma_t \leq \frac{c_0 \gamma_b}{c_t} \text{ and } \gamma_{t-1} \leq \gamma_t \text{ and } \gamma_t^2 \|\nabla f_{S_t}(x^t)\|^2 \leq \frac{\gamma_t}{c_t}[f_{S_t}(x^t) - \ell_{S_t}^*], \quad (3)$$

*where $L_{\max} = \max_{i \in [n]}\{L_i\}$.*

**Lemma C.5** (Lemma 12, (Jiang and Stich, 2023)). *For any non-negative sequence $a_0, \ldots, a_T$, the following holds:*

$$\sqrt{\sum_{t=0}^{T} a_t} \leq \sum_{t=0}^{T} \frac{a_t}{\sqrt{\sum_{s=0}^{t} a_s}} \leq 2\sqrt{\sum_{t=0}^{T} a_t}. \quad (4)$$

*If $a_0 \geq 1$, then the following holds:*

$$\sum_{t=0}^{T} \frac{a_t}{\sqrt{\sum_{s=0}^{t} a_s}} \leq \log\left(\sum_{t=0}^{T} a_t\right) + 1. \quad (5)$$

**Lemma C.6** (Lemma 13, (Jiang and Stich, 2023))**.** *If $x^2 \leq a(x + b)$ for $a \geq 0$ and $b \geq 0$, then it holds that $x \leq a + \sqrt{ab}$.*

**Lemma C.7** (Lemma 16, (Jiang and Stich, 2023))**.** *Suppose each $f_i$ is $L_i$-smooth, then the stepsize of AdaSPS satisfies*

$$\frac{1}{2cL_{\max}} \frac{1}{\sqrt{\sum_{s=0}^{t} f_{S_s}(x^s) - \ell_{S_s}^*}} \leq \eta_t \leq \frac{f_{S_t}(x^t) - \ell_{S_t}^*}{c\|\nabla f_{S_t}(x^t)\|^2} \frac{1}{\sqrt{\sum_{s=0}^{t} f_{S_s}(x^s) - \ell_{S_s}^*}}, \tag{6}$$

*where $L_{\max} = \max_{i \in [n]} \{L_i\}$.*

### C.3 ITERATE MOVING AVERAGE

As mentioned in the main paper, the iterates of the SHB method are equal to the 2.3 method. The following proposition is from Sebbouh et al. (2021). We include the proof for completeness.

**Proposition C.8** (Proposition 1.6, (Sebbouh et al., 2021))**.** *Let $\eta_t > 0$ and $\lambda_t \geq 0$. Consider the Iterate Moving Average (IMA) method given by the following update rule:*

$$z^{t+1} = z^t - \eta_t \nabla f_{S_t}(x^t) \tag{7}$$

$$x^{t+1} = \frac{\lambda_{t+1}}{\lambda_{t+1} + 1} x^t + \frac{1}{\lambda_{t+1} + 1} z^{t+1}, \tag{8}$$

*where $z^0 = x^0$. If the following equations hold for any $t \in \mathbb{N}$*

$$1 + \lambda_{t+1} = \frac{\lambda_t}{\beta_t} \tag{9}$$

$$\eta_t = (1 + \lambda_{t+1})\gamma_t, \tag{10}$$

*then the $x_t$ iterates of the 2.3 method are equal to the $x_t$ iterates produced by the SHB method.*

*Proof.* We will start from the 2.3 update rule and we will prove that its $(x^t)$ iterates satisfy the SHB update rule. Suppose that we have the $(z^{t+1})$ and $(x^{t+1})$ iterates as well as the $(\lambda_t)$ and $(\eta_t)$ sequences. Define the sequences $(\beta_t)$ and $(\gamma_t)$ using eqs. (9) and (10). We will show that $(x^t)$ iterates satisfy the SHB update rule. Solving for $z^{t+1}$ in eq. (8) we get

$$z^{t+1} := x^{t+1} + \lambda_{t+1}(x^{t+1} - x^t) \tag{11}$$

Now substituting eqs. (7) and (11) in eq. (8) we have

$$
\begin{aligned}
x^{t+1} &= \frac{\lambda_{t+1}}{\lambda_{t+1} + 1} x^t + \frac{1}{\lambda_{t+1} + 1} z^{t+1} \\
&\overset{(7)}{=} \frac{\lambda_{t+1}}{\lambda_{t+1} + 1} x^t + \frac{1}{\lambda_{t+1} + 1} \left( z^t - \eta_t \nabla f_{S_t}(x^t) \right) \\
&\overset{(11)}{=} \frac{\lambda_{t+1}}{\lambda_{t+1} + 1} x^t + \frac{1}{\lambda_{t+1} + 1} \left( x^t + \lambda_t(x^t - x^{t-1}) - \eta_t \nabla f_{S_t}(x^t) \right) \\
&= x^t - \frac{\eta_t}{\lambda_{t+1} + 1} \nabla f_{S_t}(x^t) + \frac{\lambda_t}{\lambda_{+t+1} + 1}(x^t - x^{t-1}) \\
&\overset{(9,10)}{=} x^t - \gamma_t \nabla f_{S_t}(x^t) + \beta_t(x^t - x^{t-1}),
\end{aligned}
$$

as wanted. $\square$

The above proposition essentially states that momentum is a convex combination of SGD under a certain transformation. In this paper, we work only with a constant momentum coefficient, i.e., $\beta_t = \beta$. For this case, we also assume constant $\lambda_t$, and we have Proposition 2.1 and Corollary 2.2.

*Proof of Proposition 2.1.* Setting $\lambda_t = \lambda$ in Equation (9) we have

$$1 + \lambda = \frac{\lambda}{\beta_t} \Rightarrow \beta_t = \frac{\lambda}{1 + \lambda} =: \beta.$$

Then solving for $\lambda$ we get $\lambda = \frac{\beta}{1-\beta}$. Hence

$$\begin{aligned}
\gamma_t &= \frac{\eta_t}{1 + \lambda} \\
&= \frac{\eta_t}{1 + \frac{\beta}{1-\beta}} \\
&= (1 - \beta)\eta_t
\end{aligned}$$

□

*Proof of Corollary 2.2.*
- Let $\eta_t = \text{SPS}_{\max}$. Then

$$\begin{aligned}
\gamma_t &= (1 - \beta)\eta_t \\
&= (1 - \beta)\min\left\{\frac{f_{S_t}(x^t) - \ell_{S_t}^*}{c\|\nabla f_{S_t}(x^t)\|^2}, \eta_b\right\} \\
&= (1 - \beta)\min\left\{\frac{f_{S_t}(x^t) - \ell_{S_t}^*}{c\|\nabla f_{S_t}(x^t)\|^2}, \gamma_b\right\}.
\end{aligned}$$

Here $\gamma_b = \eta_b$.

- Let $\eta_t = \text{DecSPS}$. Then

$$\begin{aligned}
\gamma_t &= (1 - \beta)\eta_t \\
&= (1 - \beta)\min\left\{\frac{f_{S_t}(x^t) - \ell_{S_t}^*}{c_t\|\nabla f_{S_t}(x^t)\|^2}, \frac{\eta_{t-1}c_{t-1}}{c_t}\right\} \\
&= \min\left\{\frac{(1 - \beta)[f_{S_t}(x^t) - \ell_{S_t}^*]}{c_t\|\nabla f_{S_t}(x^t)\|^2}, \frac{(1 - \beta)\eta_{t-1}c_{t-1}}{c_t}\right\} \\
&= \min\left\{\frac{(1 - \beta)[f_{S_t}(x^t) - \ell_{S_t}^*]}{c_t\|\nabla f_{S_t}(x^t)\|^2}, \frac{\gamma_{t-1}c_{t-1}}{c_t}\right\},
\end{aligned}$$

since $\gamma_{t-1} = (1 - \beta)\eta_{t-1}$ from Proposition 2.1.

- Let $\eta_t = \text{AdaSPS}$. Then

$$\begin{aligned}
\gamma_t &= (1 - \beta)\eta_t \\
&= (1 - \beta)\min\left\{\frac{f_{S_t}(x^t) - \ell_{S_t}^*}{c\|\nabla f_{S_t}(x^t)\|^2\sqrt{\sum_{s=0}^{t} f_{S_s}(x^s) - \ell_{S_s}^*}}, \eta_{t-1}\right\} \\
&= \min\left\{\frac{(1 - \beta)[f_{S_t}(x^t) - \ell_{S_t}^*]}{c\|\nabla f_{S_t}(x^t)\|^2\sqrt{\sum_{s=0}^{t} f_{S_s}(x^s) - \ell_{S_s}^*}}, (1 - \beta)\eta_{t-1}\right\} \\
&= \min\left\{\frac{(1 - \beta)[f_{S_t}(x^t) - \ell_{S_t}^*]}{c\|\nabla f_{S_t}(x^t)\|^2\sqrt{\sum_{s=0}^{t} f_{S_s}(x^s) - \ell_{S_s}^*}}, \gamma_{t-1}\right\}.
\end{aligned}$$

□

## C.4 CONNECTION OF MomSPS$_{\max}$ AND SPS$_{\max}$

In this section, we explore the connection between the MomSPS$_{\max}$ and the SPS$_{\max}$ step-sizes. In particular, one can view the MomSPS$_{\max}$ step-size as the SPS$_{\max}$ step-size with $c = 1/(1 - $

$\beta$). Here, we explain why these are two different step-sizes. Recall from Proposition 2.1 and Corollary 2.2 that our proposed step-sizes for the SHB setting are scaled versions of Polyak step-sizes for SGD. In particular, let's assume that

$$\gamma_t^{SPSmax} = \min\left\{\frac{f_{S_t}(x^t) - f_{S_t}^*}{c_{SPSmax}\|\nabla f_{S_t}(x^t)\|^2}, \gamma_b^{SPSmax}\right\}$$

and

$$\gamma_t^{MomSPSmax} = (1 - \beta)\min\left\{\frac{f_{S_t}(x^t) - f_{S_t}^*}{c_{MomSPSmax}\|\nabla f_{S_t}(x^t)\|^2}, \gamma_b^{MomSPSmax}\right\}$$

so we have $\gamma_{SPSmax}^t = \gamma_{MomSPSmax}^t$ if and only if

$$\frac{1 - \beta}{c_{MomSPSmax}} = \frac{1}{c_{SPSmax}} \text{ and } \gamma_b^{SPSmax} = (1 - \beta)\gamma_b^{MomSPSmax}.$$

Without loss of generalization, we can assume that $c_{MomSPSmax} = 1$ (this is also what Theorem 3.2 guarantees). In this case we get $\beta = 1 - \frac{1}{c_{SPSmax}}$. Hence, we can restate Theorem 3.2 as follows:

---

**Theorem C.9.** *Assume that each $f_i$ is convex and $L_i$-smooth. Then, the iterates of SHB with* $SPS_{\max}$ *with* $c \in \left[1, \frac{2\gamma_b - 2\alpha}{2\gamma_b - \alpha}\right)$ *and* $\beta = 1 - \frac{1}{c}$ *where* $\alpha = \min\left\{\frac{1}{2L_{\max}}, \gamma_b\right\}$ *and* $L_{\max} = \max_{i\in[n]}\{L_i\}$, *converge as*

$$\mathbb{E}[f(\overline{x}^T) - f(x^*)] \leq \frac{C_1\|x^0 - x^*\|^2}{T} + C_2\sigma^2,$$

*where* $\overline{x}^T = \frac{1}{T}\sum_{t=0}^{T-1} x^t$ *is the Cesaro average and the constants* $C_1 = \frac{1-\beta}{\alpha\beta + \alpha - 2\beta\gamma_b}$ *and* $C_2 = \frac{2\gamma_b - \alpha\beta - \alpha}{\alpha\beta + \alpha - 2\beta\gamma_b}$.

---

This means that MomSPS$_{\max}$ can be viewed as a special case of SPS$_{\max}$. However, note that the update rules are different since in SHB we have the extra term $+\beta(x^t - x^{t-1})$ so a different analysis is needed.

The same line of thought can be applied to other works related to HB. For example, in Ghadimi et al. (2015), the first global guarantees for HB were established. The authors show that for a convex and $L$-smooth function $f$ HB with any $\beta \in [0, 1)$ and $0 < \gamma < \frac{2(1-\beta)}{L}$ converges with a rate of $O(1/T)$. Also, recall that GD for convex and $L$-smooth function converges for $0 < \gamma < \frac{2}{L}$ with rate $O(1/T)$. Hence, we can restate the result of Ghadimi et al. (2015) as follows: Let $f$ be a convex and $L$-smooth function. Then HB with $\gamma = \frac{c}{L}$, where $c \in (0, 2)$, and $\beta = 1 - \frac{c}{2}$ converges with rate $O(1/T)$.

Similarly, for SGD in the convex and smooth setting, we are guaranteed convergence $O(1/T)$ to a neighborhood of the solution when $\gamma \leq \frac{1}{L}$. In Liu et al. (2020), the authors guarantee that SHB with any $\beta \in (0, 1)$ and $\gamma \leq \frac{(1-\beta)^2}{L}\min\left\{\frac{1}{4-\beta+\beta^2}, \frac{1}{2\sqrt{2\beta^2+2\beta}}\right\}$ converges with rate $O(1/T)$ to a neighborhood. This can be restated as follows: SHB with $\gamma = \frac{c}{L}$, where $c \in (0, 1)$, and $\beta$ given by the solution of $c = \min\left\{\frac{(1-\beta)^2}{4-\beta+\beta^2}, \frac{(1-\beta)^2}{2\sqrt{2\beta^2+2\beta}}\right\}$, converges with rate $O(1/T)$ to a neighborhood.

# D    Proofs of the main results

In this section we present the proofs of the main theoretical results presented in the main paper, i.e., the convergence analysis of SHB with MomSPS$_\text{max}$, MomDecSPS and MomAdaSPS for convex and smooth functions $f_i$ and $f$ of Problem 1. The idea for all the proofs is that we firstly show the corresponding result in the 2.3 setting, and then we transfer it to the SHB setting with the help of Proposition 2.1.

Compared to the analysis of SGD with SPS, the SHB update rule requires taking into consideration extra terms related to the previous iterate $x^{t-1}$. This needs new algebraic tricks in order to use convexity, and after that, one needs to deal with the two quantities $f(x^t)$ and $f(x^{t-1})$ at the same time. We deal with this using the 2.3 framework, similarly with Sebbouh et al. (2021). However, we highlight that we finish our proofs with a different approach. More specifically, in the proof of Theorem G.1 in Sebbouh et al. (2021) the authors choose the momentum coefficients $\lambda_t$ in such a way that the main sum telescopes and the final result only has the expectation of the last iterate. In our proof, we make no such simplifications, and instead, we use Jensen's inequality to finish. This approach allows us to have more freedom in the selection of $\lambda_t$. This is more prominent in the proofs of the decreasing variants.

Compared to the classical analysis of constant step-size SHB, the use of SPS requires an adaptive step-size that uses the loss and stochastic gradient estimates at an iterate, resulting in correlations of the step-size with the gradient, i.e. we might have $\mathbb{E}[\gamma_t \nabla f_i(x^t)] \neq \gamma_t \mathbb{E}[\nabla f_i(x^t)]$. One of the technical challenges in the proofs is to carefully analyze the SHB iterates, taking these correlations into account. Moreover, since we try to be adaptive to the Lipschitz constant, we can not use any standard descent lemmas (implied by the smoothness and the SHB update) that are used in the classical analysis of SGD and constant step-size SHB. This makes the convex proof more challenging than the standard analysis of SHB.

## D.1    Proof of Theorem 3.2

First, we state and prove the corresponding theorem for SPS$_\text{max}$ in the 2.3 setting.

**Theorem D.1.** *Assume that each $f_i$ is convex and $L_i$-smooth. Then, the iterates of 2.3 with*

$$\eta_t = \min\left\{ \frac{f_{S_t}(x^t) - \ell^*_{S_t}}{\|\nabla f_{S_t}(x^t)\|^2}, \eta_b \right\} \text{ and } \lambda_t = \lambda \in \left[0, \frac{\alpha}{2(\eta_b - \alpha)}\right),$$

*where $\alpha = \min\left\{\frac{1}{2L_\text{max}}, \eta_b\right\}$ and $L_\text{max} = \max_{i \in [n]}\{L_i\}$, converge as*

$$\mathbb{E}[f(\overline{x}^T) - f(x^*)] \leq \frac{\|x^0 - x^*\|^2}{(2\alpha\lambda + \alpha - 2\eta_b\lambda)T} + \frac{(2\eta_b + 2\eta_b\lambda - 2\alpha\lambda - \alpha)\sigma^2}{2\alpha\lambda + \alpha - 2\eta_b\lambda},$$

*where $\overline{x}^T = \frac{1}{T}\sum_{t=0}^{T-1} x^t$ and $\sigma^2 = \mathbb{E}[f_i(x^*) - f_i^*]$.*

*Proof.* We have

$$\|z^{t+1} - x^*\|^2 = \|z^t - x^*\|^2 - 2\eta_t\langle\nabla f_{S_t}(x^t), z^t - x^*\rangle + \eta_t^2\|\nabla f_{S_t}(x^t)\|^2$$

$$\overset{(11)}{=} \|z^t - x^*\|^2 - 2\eta_t\langle\nabla f_{S_t}(x^t), x^t - x^*\rangle - 2\eta_t\lambda_t\langle\nabla f_{S_t}(x^t), x^t - x^{t-1}\rangle$$
$$+ \eta_t^2\|\nabla f_{S_t}(x^t)\|^2$$

$$\overset{\text{convexity}}{\leq} \|z^t - x^*\|^2 - 2\eta_t[f_{S_t}(x^t) - f_{S_t}(x^*)] - 2\eta_t\lambda_t[f_{S_t}(x^t) - f_{S_t}(x^{t-1})]$$
$$+ \eta_t^2\|\nabla f_{S_t}(x^t)\|^2$$

$$\overset{(2)}{\leq} \|z^t - x^*\|^2 - 2\eta_t[f_{S_t}(x^t) - f_{S_t}(x^*)] - 2\eta_t\lambda_t[f_{S_t}(x^t) - f_{S_t}(x^{t-1})]$$
$$+ \eta_t[f_{S_t}(x^t) - \ell^*_{S_t}]$$

$$= \|z^t - x^*\|^2 - 2\eta_t[f_{S_t}(x^t) - \ell^*_{S_t}] + 2\eta_t[f_{S_t}(x^*) - \ell^*_{S_t}] - 2\eta_t\lambda_t[f_{S_t}(x^t) - \ell^*_{S_t}]$$

$$+ 2\eta_t \lambda_t [f_{S_t}(x^{t-1}) - \ell^*_{S_t}] + \eta_t [f_{S_t}(x^t) - \ell^*_{S_t}]$$

$$= \|z^t - x^*\|^2 - 2\eta_t \left(1 + \lambda_t - \frac{1}{2}\right) [f_{S_t}(x^t) - \ell^*_{S_t}] + 2\eta_t \lambda_t [f_{S_t}(x^{t-1}) - \ell^*_{S_t}]$$

$$+ 2\eta_t [f_{S_t}(x^*) - \ell^*_{S_t}]$$

$$\overset{(2)}{\leq} \|z^t - x^*\|^2 - 2\alpha \left(\lambda_t + \frac{1}{2}\right) [f_{S_t}(x^t) - \ell^*_{S_t}] + 2\eta_b \lambda_t [f_{S_t}(x^{t-1}) - \ell^*_{S_t}]$$

$$+ 2\eta_b [f_{S_t}(x^*) - \ell^*_{S_t}]$$

$$\leq \|z^t - x^*\|^2 - 2\alpha \left(\lambda_t + \frac{1}{2}\right) [f_{S_t}(x^t) - f_{S_t}(x^*)] - 2\alpha \left(\lambda_t + \frac{1}{2}\right) [f_{S_t}(x^*) - \ell^*_{S_t}]$$

$$+ 2\eta_b \lambda_t [f_{S_t}(x^{t-1}) - f_{S_t}(x^*)] + 2\eta_b \lambda_t [f_{S_t}(x^*) - \ell^*_{S_t}] + 2\eta_b [f_{S_t}(x^*) - \ell^*_{S_t}].$$

Now take expectation conditional on $x^t$ to get

$$\mathbb{E}_{S_t} \|z^{t+1} - x^*\|^2 \leq \mathbb{E}_{S_t} \|z^t - x^*\|^2 - 2\alpha \left(\lambda_t + \frac{1}{2}\right) \mathbb{E}_{S_t}[f_{S_t}(x^t) - f_{S_t}(x^*)]$$

$$- 2\alpha \left(\lambda_t + \frac{1}{2}\right) \mathbb{E}_{S_t}[f_{S_t}(x^*) - \ell^*_{S_t}] + 2\eta_b \lambda_t \mathbb{E}_{S_t}[f_{S_t}(x^{t-1}) - f_{S_t}(x^*)]$$

$$+ 2\eta_b \lambda_t \mathbb{E}_{S_t}[f_{S_t}(x^*) - \ell^*_{S_t}] + 2\eta_b \mathbb{E}_{S_t}[f_{S_t}(x^*) - \ell^*_{S_t}]$$

$$= \|z^t - x^*\|^2 - 2\alpha \left(\lambda_t + \frac{1}{2}\right) [f(x^t) - f(x^*)] - 2\alpha \left(\lambda_t + \frac{1}{2}\right) \sigma^2$$

$$+ 2\eta_b \lambda_t [f(x^{t-1}) - f(x^*)] + 2\eta_b \lambda_t \sigma^2 + 2\eta_b \sigma^2$$

$$= \|z^t - x^*\|^2 - 2\alpha \left(\lambda_t + \frac{1}{2}\right) [f(x^t) - f(x^*)] + 2\eta_b \lambda_t [f(x^{t-1}) - f(x^*)]$$

$$+ 2 \left(\eta_b (1 + \lambda_t) - \alpha \left(\lambda_t + \frac{1}{2}\right)\right) \sigma^2. \tag{12}$$

Note that in eq. (12) the coefficient $2\left(\eta_b (1 + \lambda_t) - \alpha \left(\lambda_t + \frac{1}{2}\right)\right)$ of $\sigma^2$ is positive since $\eta_b \geq \alpha$. Taking expectation again and using the tower property we have

$$2\alpha \left(\lambda_t + \frac{1}{2}\right) \mathbb{E}[f(x^t) - f(x^*)] - 2\eta_b \lambda_t \mathbb{E}[f(x^{t-1}) - f(x^*)]$$

$$\leq \mathbb{E} \|z^t - x^*\|^2 - \mathbb{E} \|z^{t+1} - x^*\|^2 + 2 \left(\eta_b (1 + \lambda_t) - \alpha \left(\lambda_t + \frac{1}{2}\right)\right) \sigma^2. \tag{13}$$

Summing eq. (13) for $t = 0, \ldots, T - 1$ and telescoping we get

$$\sum_{t=0}^{T-1} \left[ 2\alpha \left(\lambda_t + \frac{1}{2}\right) \mathbb{E}[f(x^t) - f(x^*)] - 2\eta_b \lambda_t \mathbb{E}[f(x^{t-1}) - f(x^*)] \right]$$

$$\leq \sum_{t=0}^{T-1} \left[ \mathbb{E} \|z^t - x^*\|^2 - \mathbb{E} \|z^{t+1} - x^*\|^2 \right] + \sum_{t=0}^{T-1} 2 \left(\eta_b (1 + \lambda_t) - \alpha \left(\lambda_t + \frac{1}{2}\right)\right) \sigma^2$$

$$= \mathbb{E} \|z^0 - x^*\|^2 - \mathbb{E} \|z^T - x^*\|^2 + \sum_{t=0}^{T-1} 2 \left(\eta_b (1 + \lambda_t) - \alpha \left(\lambda_t + \frac{1}{2}\right)\right) \sigma^2$$

$$\overset{z^0 = x^0}{\leq} \|x^0 - x^*\|^2 + \sum_{t=0}^{T-1} 2 \left(\eta_b (1 + \lambda_t) - \alpha \left(\lambda_t + \frac{1}{2}\right)\right) \sigma^2. \tag{14}$$

The LHS of eq. (14) can be bounded as follows (using $x^{-1} = x^0$)

$$\sum_{t=0}^{T-1} \left[ 2\alpha \left(\lambda_t + \frac{1}{2}\right) \mathbb{E}[f(x^t) - f(x^*)] - 2\eta_b \lambda_t \mathbb{E}[f(x^{t-1}) - f(x^*)] \right]$$

$$= \sum_{t=0}^{T-1} 2\alpha \left( \lambda_t + \frac{1}{2} \right) \mathbb{E}[f(x^t) - f(x^*)] - \sum_{t=0}^{T-1} 2\eta_b \lambda_t \mathbb{E}[f(x^{t-1}) - f(x^*)]$$

$$= \sum_{t=0}^{T-1} 2\alpha \left( \lambda_t + \frac{1}{2} \right) \mathbb{E}[f(x^t) - f(x^*)] - \sum_{t=0}^{T-2} 2\eta_b \lambda_{t+1} \mathbb{E}[f(x^t) - f(x^*)]$$

$$= 2\alpha \left( \lambda_{T-1} + \frac{1}{2} \right) \mathbb{E}[f(x^{T-1}) - f(x^*)] + \sum_{t=0}^{T-2} 2\alpha \left( \lambda_t + \frac{1}{2} \right) \mathbb{E}[f(x^t) - f(x^*)]$$

$$- \sum_{t=0}^{T-2} 2\eta_b \lambda_{t+1} \mathbb{E}[f(x^t) - f(x^*)]$$

$$= 2\alpha \left( \lambda_{T-1} + \frac{1}{2} \right) \mathbb{E}[f(x^{T-1}) - f(x^*)] + \sum_{t=0}^{T-2} \left[ 2\alpha \left( \lambda_t + \frac{1}{2} \right) - 2\eta_b \lambda_{t+1} \right] \mathbb{E}[f(x^t) - f(x^*)]$$

$$\geq \left[ 2\alpha \left( \lambda_{T-1} + \frac{1}{2} \right) - 2\eta_b \lambda_T \right] \mathbb{E}[f(x^{T-1}) - f(x^*)]$$

$$+ \sum_{t=0}^{T-2} \left[ 2\alpha \left( \lambda_t + \frac{1}{2} \right) - 2\eta_b \lambda_{t+1} \right] \mathbb{E}[f(x^t) - f(x^*)]$$

$$= \sum_{t=0}^{T-1} \left[ 2\alpha \left( \lambda_t + \frac{1}{2} \right) - 2\eta_b \lambda_{t+1} \right] \mathbb{E}[f(x^t) - f(x^*)]. \tag{15}$$

Combining eq. (14) and eq. (15) we have

$$\sum_{t=0}^{T-1} \left[ 2\alpha \left( \lambda_t + \frac{1}{2} \right) - 2\eta_b \lambda_{t+1} \right] \mathbb{E}[f(x^t) - f(x^*)]$$

$$\leq \|x^0 - x^*\|^2 + \sum_{t=0}^{T-1} 2 \left( \eta_b (1 + \lambda_t) - \alpha \left( \lambda_t + \frac{1}{2} \right) \right) \sigma^2. \tag{16}$$

In eq. (16), setting $\lambda_t = \lambda, \forall t > 0$ with $0 \leq \lambda < \frac{\alpha}{2(\eta_b - \alpha)}$ (in order to ensure that quantity $2\alpha \left( \lambda + \frac{1}{2} \right) - 2\eta_b \lambda$ in the LHS is positive) we get

$$\left[ 2\alpha \left( \lambda + \frac{1}{2} \right) - 2\eta_b \lambda \right] \sum_{t=0}^{T-1} \mathbb{E}[f(x^t) - f(x^*)] \leq \|x^0 - x^*\|^2 + 2T \left( \eta_b (1 + \lambda) - \alpha \left( \lambda + \frac{1}{2} \right) \right) \sigma^2. \tag{17}$$

Now dividing eq. (17) by $T \left[ 2\alpha \left( \lambda + \frac{1}{2} \right) - 2\eta_b \lambda \right]$ and using Jensen's inequality we get

$$\mathbb{E}[f(\bar{x}^T) - f(x^*)] \leq \frac{1}{T} \sum_{t=0}^{T-1} \mathbb{E}[f(x^t) - f(x^*)]$$

$$\leq \frac{\|x^0 - x^*\|^2 + T \left( 2\eta_b + 2\eta_b \lambda - 2\alpha\lambda - \alpha \right) \sigma^2}{(2\alpha\lambda + \alpha - 2\eta_b\lambda)T}$$

$$= \frac{\|x^0 - x^*\|^2}{(2\alpha\lambda + \alpha - 2\eta_b\lambda)T} + \frac{2\eta_b + 2\eta_b\lambda - 2\alpha\lambda - \alpha}{2\alpha\lambda + \alpha - 2\eta_b\lambda} \sigma^2,$$

as wanted. □

Now the proof of Theorem 3.2 follows immediately from Theorem D.1 setting $\lambda = \frac{\beta}{1-\beta}$ (from Proposition 2.1) and $\gamma_b = \eta_b$ (from Corollary 2.2). The bound on $\beta = \frac{\lambda}{1+\lambda}$ follows from the bound on $\lambda$ and from the fact that the function $f(x) = \frac{x}{1+x}$ is strictly increasing for $x \geq 0$.

### D.2 PROOF OF THEOREM 3.6

Here we state and prove the corresponding theorem for DecSPS in the 2.3 setting.

> **Theorem D.2.** *Assume that each $f_i$ is convex and $L_i$-smooth. Then, the iterates of 2.3 with*
>
> $$\eta_t = \frac{1}{c_t} \min \left\{ \frac{f_{S_t}(x^t) - \ell^*_{S_t}}{\|\nabla f_{S_t}(x^t)\|^2}, c_{t-1}\eta_{t-1} \right\} \text{ and } \lambda_t \geq 0,$$
>
> *where $(c_t)$ is an increasing sequences with $c_t \geq 1$, $c_{-1} = c_0$, $\eta_{-1} = \eta_b$ and $(\lambda_t)$ is a decreasing sequence, converge as*
>
> $$\mathbb{E}[f(\overline{x}^T) - f(x^*)] \leq \frac{2\lambda_1[f(x^0) - f(x^*)]}{T} + \frac{c_{T-1}D_z^2}{T\alpha} + \frac{\sigma^2}{T}\sum_{t=0}^{T-1}\frac{1}{c_t},$$
>
> *where $\overline{x}^T = \frac{1}{T}\sum_{t=0}^{T-1} x^t$, $\alpha = \min\left\{\frac{1}{2L}, \eta_b c_0\right\}$ and $D_z^2 = \max_{t\in[T]} \|z^t - x^*\|^2$.*

*Proof.* We have

$$\|z^{t+1} - x^*\|^2 \overset{(7)}{=} \|z^t - x^*\|^2 - 2\eta_t\langle\nabla f_{S_t}(x^t), z^t - x^*\rangle + \eta_t^2\|\nabla f_{S_t}(x^t)\|^2.$$

Rearranging and dividing by $\eta_t > 0$ we get

$$2\langle\nabla f_{S_t}(x^t), z^t - x^*\rangle \leq \frac{\|z^t - x^*\|^2}{\eta_t} - \frac{\|z^{t+1} - x^*\|^2}{\eta_t} + \eta_t\|\nabla f_{S_t}(x^t)\|^2.$$

Summing the above for $t = 0, \ldots, T - 1$ and telescoping we have

$$2\sum_{t=0}^{T-1}\langle\nabla f_{S_t}(x^t), z^t - x^*\rangle$$

$$\leq \sum_{t=0}^{T-1}\left[\frac{\|z^t - x^*\|^2}{\eta_t} - \frac{\|z^{t+1} - x^*\|^2}{\eta_t} + \eta_t\|\nabla f_{S_t}(x^t)\|^2\right]$$

$$= \sum_{t=0}^{T-1}\frac{\|z^t - x^*\|^2}{\eta_t} - \sum_{t=0}^{T-1}\frac{\|z^{t+1} - x^*\|^2}{\eta_t} + \sum_{t=0}^{T-1}\eta_t\|\nabla f_{S_t}(x^t)\|^2$$

$$= \frac{\|z^0 - x^*\|^2}{\eta_0} + \sum_{t=1}^{T-1}\frac{\|z^t - x^*\|^2}{\eta_t} - \sum_{t=0}^{T-1}\frac{\|z^{t+1} - x^*\|^2}{\eta_t} + \sum_{t=0}^{T-1}\eta_t\|\nabla f_{S_t}(x^t)\|^2$$

$$= \frac{\|z^0 - x^*\|^2}{\eta_0} + \sum_{t=0}^{T-2}\frac{\|z^{t+1} - x^*\|^2}{\eta_{t+1}} - \sum_{t=0}^{T-1}\frac{\|z^{t+1} - x^*\|^2}{\eta_t} + \sum_{t=0}^{T-1}\eta_t\|\nabla f_{S_t}(x^t)\|^2$$

$$= \frac{\|z^0 - x^*\|^2}{\eta_0} + \sum_{t=0}^{T-2}\frac{\|z^{t+1} - x^*\|^2}{\eta_{t+1}} - \sum_{t=0}^{T-2}\frac{\|z^{t+1} - x^*\|^2}{\eta_t} - \frac{\|z^T - x^*\|^2}{\eta_{T-1}} + \sum_{t=0}^{T-1}\eta_t\|\nabla f_{S_t}(x^t)\|^2$$

$$= \frac{\|z^0 - x^*\|^2}{\eta_0} + \sum_{t=0}^{T-2}\left(\frac{1}{\eta_{t+1}} - \frac{1}{\eta_t}\right)\|z^{t+1} - x^*\|^2 - \frac{\|z^T - x^*\|^2}{\eta_{T-1}} + \sum_{t=0}^{T-1}\eta_t\|\nabla f_{S_t}(x^t)\|^2$$

$$\leq \left(\frac{1}{\eta_0} + \sum_{t=0}^{T-2}\left[\frac{1}{\eta_{t+1}} - \frac{1}{\eta_t}\right]\right)D_z^2 + \sum_{t=0}^{T-1}\eta_t\|\nabla f_{S_t}(x^t)\|^2$$

$$= \frac{D_z^2}{\eta_{T-1}} + \sum_{t=0}^{T-1}\eta_t\|\nabla f_{S_t}(x^t)\|^2. \tag{18}$$

Taking expectation in eq. (18) we have

$$2\sum_{t=0}^{T-1}\mathbb{E}\left[\langle\nabla f(x^t), z^t - x^*\rangle\right] \leq \mathbb{E}\left[\frac{D_z^2}{\eta_{T-1}} + \sum_{t=0}^{T-1}\eta_t\|\nabla f_{S_t}(x^t)\|^2\right]. \tag{19}$$

Now the LHS of eq. (19) can be bounded as follows

$$2 \sum_{t=0}^{T-1} \mathbb{E} \left[ \langle \nabla f(x^t), z^t - x^* \rangle \right]$$

$$\overset{(z^0 = x^0)}{=} 2 \mathbb{E} \left[ \langle \nabla f(x^0), x^0 - x^* \rangle \right] + 2 \sum_{t=1}^{T-1} \mathbb{E} \left[ \langle \nabla f(x^t), z^t - x^* \rangle \right]$$

$$\overset{(11)}{=} 2 \mathbb{E} \left[ \langle \nabla f(x^0), x^0 - x^* \rangle \right] + \sum_{t=1}^{T-1} \mathbb{E} \left[ 2 \langle \nabla f(x^t), x^t - x^* \rangle + 2 \lambda_t \langle \nabla f(x^t), x^t - x^{t-1} \rangle \right]$$

$$\overset{\text{convexity}}{\geq} 2 \mathbb{E} \left[ f(x^0) - f(x^*) \right] + \sum_{t=1}^{T-1} \mathbb{E} \left[ 2(f(x^t) - f(x^*)) + 2 \lambda_t (f(x^t) - f(x^{t-1})) \right]$$

$$= 2 \mathbb{E} \left[ f(x^0) - f(x^*) \right] + \sum_{t=1}^{T-1} \left[ 2(1 + \lambda_t) \mathbb{E}[f(x^t) - f(x^*)] - 2 \lambda_t \mathbb{E}[f(x^{t-1}) - f(x^*)] \right]$$

$$= 2 \mathbb{E} \left[ f(x^0) - f(x^*) \right] + \sum_{t=1}^{T-1} 2(1 + \lambda_t) \mathbb{E}[f(x^t) - f(x^*)] - \sum_{t=1}^{T-1} 2 \lambda_t \mathbb{E}[f(x^{t-1}) - f(x^*)]$$

$$= 2 \mathbb{E} \left[ f(x^0) - f(x^*) \right] + \sum_{t=1}^{T-1} 2(1 + \lambda_t) \mathbb{E}[f(x^t) - f(x^*)] - \sum_{t=0}^{T-2} 2 \lambda_{t+1} \mathbb{E}[f(x^t) - f(x^*)]$$

$$= 2 \mathbb{E} \left[ f(x^0) - f(x^*) \right] + 2 \sum_{t=1}^{T-1} \mathbb{E}[f(x^t) - f(x^*)] + \sum_{t=1}^{T-1} 2 \lambda_t \mathbb{E}[f(x^t) - f(x^*)]$$

$$\quad - \sum_{t=0}^{T-2} 2 \lambda_{t+1} \mathbb{E}[f(x^t) - f(x^*)]$$

$$= 2 \mathbb{E} \left[ f(x^0) - f(x^*) \right] + 2 \sum_{t=1}^{T-1} \mathbb{E}[f(x^t) - f(x^*)] + 2 \lambda_{T-1} \mathbb{E}[f(x^{T-1}) - f(x^*)]$$

$$\quad + \sum_{t=1}^{T-2} 2 \lambda_t \mathbb{E}[f(x^t) - f(x^*)] - \sum_{t=1}^{T-2} 2 \lambda_{t+1} \mathbb{E}[f(x^t) - f(x^*)] - 2 \lambda_1 \mathbb{E}[f(x^0) - f(x^*)]$$

$$= 2 \sum_{t=0}^{T-1} \mathbb{E}[f(x^t) - f(x^*)] + 2 \lambda_{T-1} \mathbb{E}[f(x^{T-1}) - f(x^*)]$$

$$\quad + 2 \sum_{t=1}^{T-2} (\lambda_t - \lambda_{t+1}) \mathbb{E}[f(x^t) - f(x^*)] - 2 \lambda_1 \mathbb{E}[f(x^0) - f(x^*)]$$

$$\geq 2 \sum_{t=0}^{T-1} \mathbb{E}[f(x^t) - f(x^*)] - 2 \lambda_1 [f(x^0) - f(x^*)], \tag{20}$$

using the fact that $(\lambda_t)$ is decreasing and the fact that $\mathbb{E}[f(x^t) - f(x^*)] \geq 0$. Combining eq. (19) and eq. (20) we have

$$2 \sum_{t=0}^{T-1} \mathbb{E}[f(x^t) - f(x^*)] - 2 \lambda_1 [f(x^0) - f(x^*)] \leq \mathbb{E} \left[ \frac{D_z^2}{\eta_{T-1}} + \sum_{t=0}^{T-1} \mathbb{E} \, \eta_t \| \nabla f_{S_t}(x^t) \|^2 \right]. \tag{21}$$

Now for the RHS of eq. (21) we have

$$\frac{D_z^2}{\eta_{T-1}} + \sum_{t=0}^{T-1} \eta_t \| \nabla f_{S_t}(x^t) \|^2 \overset{(3)}{\leq} \frac{D_z^2}{\eta_{T-1}} + \sum_{t=0}^{T-1} \frac{1}{c_t} [f_{S_t}(x^t) - \ell_{S_t}^*]$$

$$= \frac{D_z^2}{\eta_{T-1}} + \sum_{t=0}^{T-1} \frac{1}{c_t} [f_{S_t}(x^t) - f_{S_t}(x^*)] + \sum_{t=0}^{T-1} \frac{1}{c_t} [f_{S_t}(x^*) - \ell_{S_t}^*]$$

$$\leq \frac{c_{T-1}D_z^2}{\alpha} + \sum_{t=0}^{T-1} \frac{1}{c_t}[f_{S_t}(x^t) - f_{S_t}(x^*)] + \sum_{t=0}^{T-1} \frac{1}{c_t}[f_{S_t}(x^*) - \ell_{S_t}^*],$$

(22)

where the last inequality follows from the fact that $\eta_t$ is decreasing and $\eta_t \geq \frac{\alpha}{c_t}$ with $\alpha = \min\{\frac{1}{2L_{\max}}, \eta_b c_0\}$, from Lemma C.4. Thus, combining eq. (21) and eq. (22) we have

$$2\sum_{t=0}^{T-1} \mathbb{E}[f(x^t) - f(x^*)] - 2\lambda_1[f(x^0) - f(x^*)] \leq \frac{c_{T-1}D_z^2}{\alpha} + \sum_{t=0}^{T-1} \frac{1}{c_t}\mathbb{E}[f(x^t) - f(x^*)] + \sum_{t=0}^{T-1} \frac{\sigma^2}{c_t}.$$

(23)

Rearranging eq. (23) we get

$$\sum_{t=0}^{T-1} \left(2 - \frac{1}{c_t}\right) \mathbb{E}[f(x^t) - f(x^*)] \leq 2\lambda_1[f(x^0) - f(x^*)] + \frac{c_{T-1}D_z^2}{\alpha} + \sigma^2 \sum_{t=0}^{T-1} \frac{1}{c_t}$$

(24)

and using the fact that $2 - \frac{1}{c_t} \geq 1 > 0$ eq. (24) reduces to

$$\sum_{t=0}^{T-1} \mathbb{E}[f(x^t) - f(x^*)] \leq 2\lambda_1[f(x^0) - f(x^*)] + \frac{c_{T-1}D_z^2}{\alpha} + \sigma^2 \sum_{t=0}^{T-1} \frac{1}{c_t}$$

(25)

Dividing eq. (25) by $T$ and using Jensen's inequality we get

$$\mathbb{E}[f(\overline{x}^T) - f(x^*)] \leq \frac{1}{T}\sum_{t=0}^{T-1} \mathbb{E}[f(x^t) - f(x^*)]$$

$$\leq \frac{2\lambda_1[f(x^0) - f(x^*)]}{T} + \frac{c_{T-1}D_z^2}{T\alpha} + \frac{\sigma^2}{T}\sum_{t=0}^{T-1} \frac{1}{c_t},$$

as wanted. $\qquad\square$

If $\sigma^2 > 0$ like in the standard SGD analysis under decreasing step-sizes, the choice $c_t = \sqrt{t+1}$ leads to the asymptotic rate $O(1/\sqrt{t})$.

**Corollary D.3.** *Under the setting of Theorem D.2, for $c_t = \sqrt{t+1}$ ($c_{-1} = c_0$) we have*

$$\mathbb{E}[f(\overline{x}^T) - f(x^*)] \leq \frac{2\lambda_1[f(x^0) - f(x^*)]}{T} + \frac{D_z^2}{\alpha\sqrt{T}} + \frac{2\sigma^2}{\sqrt{T}}.$$

*Proof.* It follows from the well known inequality

$$\sum_{t=0}^{T-1} \frac{1}{\sqrt{t+1}} \leq 2\sqrt{T}.$$

$\qquad\square$

Now the proof of Theorem 3.6 follows immediately from Corollary D.3 setting $\lambda_t = \lambda = \frac{\beta}{1-\beta}$ (from Proposition 2.1). Moreover we have

$$\|z^t - x^*\|^2 \overset{(11)}{=} \|(1+\lambda)x^t - \lambda x^{t-1} - x^*\|^2$$

$$\leq (1+\lambda)\|x^t - x^*\|^2 + \lambda\|x^{t-1} - x^*\|^2$$

$$\leq (1+2\lambda)\max_t \|x^t - x^*\|^2,$$

(26)

thus $D_z^2 \leq (1+2\lambda)\max_t \|x^t - x^*\|^2 = \frac{1+\beta}{1-\beta}D^2$, where $D^2 = \max_{t\in[T]} \|x^t - x^*\|$.

### D.3 PROOF OF THEOREM 3.7

Here we state and prove the corresponding theorem for AdaSPS in the 2.3 setting.

**Theorem D.4.** *Assume that each $f_i$ is convex and $L_i$-smooth. Then, the iterates of 2.3 with*

$$\eta_t = \min\left\{\frac{f_{S_t}(x^t) - \ell_{S_t}^*}{c\|\nabla f_{S_t}(x^t)\|^2}\frac{1}{\sqrt{\sum_{s=0}^t f_{S_s}(x^s) - \ell_{S_s}^*}}, \eta_{t-1}\right\} \text{ and } \lambda_t \geq 0,$$

*where $\eta_{-1} = +\infty$, $c > 0$ and $(\lambda_t)$ a decreasing sequence, converge as*

$$\mathbb{E}[f(\overline{x}^T) - f(x^*)] \leq \frac{\tau^2}{T} + \frac{\tau\sigma}{\sqrt{T}},$$

*where $\overline{x}^T = \frac{1}{T}\sum_{t=0}^{T-1} x^t$, $\tau = \left(\lambda_1\sqrt{f(x^0) - f(x^*)} + \frac{cLD_z^2}{2} + \frac{1}{2c}\right)$ and $D_z^2 = \max_t\|z^t - x^*\|$.*

*Proof.* Using Equation (21) from the proof of Theorem 3.6, we have

$$2\sum_{t=0}^{T-1}\mathbb{E}[f(x^t) - f(x^*)] - 2\lambda_1[f(x^0) - f(x^*)] \leq \mathbb{E}\left[\frac{D_z^2}{\eta_{T-1}} + \sum_{t=0}^{T-1}\mathbb{E}\,\eta_t\|\nabla f_{S_t}(x^t)\|^2\right]. \quad (27)$$

By Lemma C.7 we have

$$\frac{D_z^2}{\eta_{T-1}} \leq cLD_z^2\sqrt{\sum_{s=0}^{T-1} f_{S_s}(x^s) - \ell_{S_s}^*}, \quad (28)$$

and by Lemma C.5 we have

$$\sum_{t=0}^{T-1}\eta_t\|\nabla f_{S_t}(x^t)\|^2 \leq \sum_{t=0}^{T-1}\frac{f_{S_t}(x^t) - \ell_{S_t}^*}{2c\sqrt{\sum_{s=0}^t f_{S_s}(x^s) - \ell_{S_s}^*}} \leq \frac{1}{c}\sqrt{\sum_{s=0}^{T-1} f_{S_s}(x^s) - \ell_{S_s}^*}. \quad (29)$$

Now combining Equations (27) to (29) and using Jensen's inequality we get

$$2\sum_{t=0}^{T-1}\mathbb{E}[f(x^t) - f(x^*)] - 2\lambda_1[f(x^0) - f(x^*)]$$

$$\leq \mathbb{E}\left[cLD_z^2\sqrt{\sum_{s=0}^{T-1} f_{S_s}(x^s) - \ell_{S_s}^*} + \frac{1}{c}\sqrt{\sum_{s=0}^{T-1} f_{S_s}(x^s) - \ell_{S_s}^*}\right]$$

$$= \left(cLD_z^2 + \frac{1}{c}\right)\mathbb{E}\left[\sqrt{\sum_{s=0}^{T-1} f_{S_s}(x^s) - \ell_{S_s}^*}\right]$$

$$= \left(cLD_z^2 + \frac{1}{c}\right)\mathbb{E}\left[\sqrt{\sum_{s=0}^{T-1}(f_{S_s}(x^s) - f_{S_s}(x^*) + f_{S_s}(x^*) - \ell_{S_s}^*)}\right]$$

$$\leq \left(cLD_z^2 + \frac{1}{c}\right)\sqrt{\sum_{s=0}^{T-1}\mathbb{E}[f(x^s) - f(x^*)] + \sigma^2}. \quad (30)$$

Rearranging eq. (30) we have

$$2\sum_{t=0}^{T-1}\mathbb{E}[f(x^t) - f(x^*)] \leq 2\lambda_1[f(x^0) - f(x^*)] + \left(cLD_z^2 + \frac{1}{c}\right)\sqrt{\sum_{s=0}^{T-1}\mathbb{E}[f(x^s) - f(x^*)] + \sigma^2}. \quad (31)$$

Now let us choose $c_q = \sqrt{f(x^0) - f(x^*)}$. Then

$$
\begin{aligned}
2\lambda_1[f(x^0) - f(x^*)] = 2\lambda_1 c_q \sqrt{f(x^0) - f(x^*)} \\
\leq 2\lambda_1 c_q \sqrt{\sum_{s=0}^{T-1} \mathbb{E}[f(x^s) - f(x^*)] + \sigma^2},
\end{aligned}
\tag{32}
$$

Combining eq. (31) and eq. (32) we get

$$
\sum_{t=0}^{T-1} \mathbb{E}[f(x^t) - f(x^*)] \leq \left( \lambda_1 c_q + \frac{cLD_z^2}{2} + \frac{1}{2c} \right) \sqrt{\sum_{s=0}^{T-1} \mathbb{E}[f(x^s) - f(x^*)] + \sigma^2}.
\tag{33}
$$

Squaring both sides of eq. (33) we have

$$
\left( \sum_{t=0}^{T-1} \mathbb{E}[f(x^t) - f(x^*)] \right)^2 \leq \tau^2 \left( \sum_{t=0}^{T-1} \mathbb{E}[f(x^t) - f(x^*)] + T\sigma^2 \right),
\tag{34}
$$

where $\tau = \left( \lambda_1 c_q + \frac{cLD_z^2}{2} + \frac{1}{2c} \right)$. Now we use Lemma C.6 in eq. (33) to get

$$
\sum_{t=0}^{T-1} \mathbb{E}[f(x^t) - f(x^*)] \leq \tau^2 + \tau\sigma\sqrt{T}.
\tag{35}
$$

Finally by Jensen's inequality and eq. (35) we have

$$
\mathbb{E}[f(\overline{x}^T) - f(x^*)] \leq \frac{\tau^2}{T} + \frac{\tau\sigma}{\sqrt{T}},
$$

as wanted. $\qquad\square$

Now the proof of Theorem 3.7 follows immediately from Theorem D.4 setting $\lambda = \frac{\beta}{1-\beta}$ (from Proposition 2.1). Moreover, using eq. (26) we have $D_z^2 \leq \frac{1+\beta}{1-\beta}D^2$.

# E  BEYOND THE BOUNDED ITERATES ASSUMPTION

The two main theorems on the convergence of MomDecSPS/MomAdaSPS require the bounded iterates assumption. As mentioned in the main paper, this is a standard assumption for several adaptive step-sizes, see: (Reddi et al., 2018; Ward et al., 2020; Orvieto et al., 2022; Jiang and Stich, 2023). However, one can artificially remove this assumption by adding a projection step in the update rule of IMA onto a compact and convex subset $\mathcal{D} \subseteq \mathbb{R}^d$ as described in this section.

Consider the *constrained* finite-sum optimization problem,

$$\min_{x \in \mathcal{D}} \left[ f(x) = \frac{1}{n} \sum_{i=1}^{n} f_i(x) \right], \tag{36}$$

where each $f_i : \mathbb{R}^d \to \mathbb{R}$ is convex, smooth, and lower bounded by $\ell_i^*$ and $\mathcal{D} \subseteq \mathbb{R}^d$ is a compact and convex subset. Let $X_{\mathcal{D}}^* \subseteq \mathcal{D}$ be the set of minimizers of (36). We assume that $X_{\mathcal{D}}^* \neq \emptyset$ and we fix $x^* \in X_{\mathcal{D}}^*$.

The new update rule of IMA, takes the following form (Projected IMA):

$$z^{t+1} = \text{proj}_{\mathcal{D}} \left[ z^t - \eta_t \nabla f_{S_t}(x^t) \right] \tag{37}$$

$$x^{t+1} = \frac{\lambda_{t+1}}{\lambda_{t+1} + 1} x^t + \frac{1}{\lambda_{t+1} + 1} z^{t+1}, \tag{38}$$

with $x^0 = z^0 \in \mathcal{D}$, where $\text{proj}_{\mathcal{D}}(x) \in \arg\min_{d \in \mathcal{D}} \|d - x\|^2$. Note that the projection step is only needed for the update of $z^{t+1}$ because $x^{t+1}$ is already a convex combination of elements in the convex set $\mathcal{D}$, namely $z^{t+1}$ and $x^t$ by induction. We highlight the fact that Projected IMA is not necessarily equivalent to Projected SHB. Now the proofs of Theorems D.2 and D.4 will go through using the non-expansiveness of the projection operator, as explained in the following lemma:

**Lemma E.1** (Non-expansiveness). *For all $x, y \in \mathbb{R}^d$ it holds*

$$\|\text{proj}_{\mathcal{D}}(x) - \text{proj}_{\mathcal{D}}(y)\| \leq \|x - y\|$$

*Proof.* See Example 8.14 and Lemma 8.16 in Garrigos and Gower (2023). □

Now we have:

**Theorem E.2** (Pojected IMA version of Theorem D.2). *Assume that each $f_i$ is convex and $L_i$-smooth. Then, the iterates of Projected IMA with*

$$\eta_t = \frac{1}{c_t} \min \left\{ \frac{f_{S_t}(x^t) - \ell_{S_t}^*}{\|\nabla f_{S_t}(x^t)\|^2}, c_{t-1} \eta_{t-1} \right\} \text{ and } \lambda_t \geq 0,$$

*where $(c_t)$ is an increasing sequences with $c_t \geq 1$, $c_{-1} = c_0$, $\eta_{-1} = \eta_b$ and $(\lambda_t)$ is a decreasing sequence, converge as*

$$\mathbb{E}[f(\overline{x}^T) - f(x^*)] \leq \frac{2\lambda_1[f(x^0) - f(x^*)]}{T} + \frac{c_{T-1}D^2}{T\alpha} + \frac{\sigma^2}{T} \sum_{t=0}^{T-1} \frac{1}{c_t},$$

*where $\overline{x}^T = \frac{1}{T} \sum_{t=0}^{T-1} x^t$, $\alpha = \min\left\{\frac{1}{2L}, \eta_b c_0\right\}$ and $D = \text{diam } \mathcal{D}$.*

*Proof.* We have

$$\|z^{t+1} - x^*\|^2 = \|\text{proj}_{\mathcal{D}}(z^t - \eta_t \nabla f_{S_t}(x^t)) - \text{proj}_{\mathcal{D}}(x^*)\|$$

$$\overset{\text{Lem. E.1}}{\leq} \|z^t - \eta_t \nabla f_{S_t}(x^t) - x^*\|$$

$$= \|z^t - x^*\|^2 - 2\eta_t \langle \nabla f_{S_t}(x^t), z^t - x^* \rangle + \eta_t^2 \|\nabla f_{S_t}(x^t)\|^2.$$

Now we continue exactly like the rest of the proof of Theorem D.2. □

**Theorem E.3** (Pojected IMA version of Theorem D.4). *Assume that each $f_i$ is convex and $L_i$-smooth. Then, the iterates of Projected IMA with*

$$\eta_t = \min\left\{\frac{f_{S_t}(x^t) - \ell_{S_t}^*}{c\|\nabla f_{S_t}(x^t)\|^2} \frac{1}{\sqrt{\sum_{s=0}^t f_{S_s}(x^s) - \ell_{S_s}^*}}, \eta_{t-1}\right\} \text{ and } \lambda_t \geq 0,$$

*where $\eta_{-1} = +\infty$, $c > 0$ and $(\lambda_t)$ a decreasing sequence, converge as*

$$\mathbb{E}[f(\overline{x}^T) - f(x^*)] \leq \frac{\tau^2}{T} + \frac{\tau\sigma}{\sqrt{T}},$$

*where $\overline{x}^T = \frac{1}{T}\sum_{t=0}^{T-1} x^t$, $\tau = \left(\lambda_1\sqrt{f(x^0) - f(x^*)} + \frac{cLD^2}{2} + \frac{1}{2c}\right)$ and $D = \text{diam } \mathcal{D}$.*

*Proof.* Same as above. □

## F  ADDITIONAL EXPERIMENTS

### F.1  EXTRA DETERMINISTIC EXPERIMENT

Here we have included an extra experiment in the deterministic setting, this one for logistic regression. We have performed a grid search to find the best $\beta$ for $\text{MomPS}_{\max}$ since no optimal choices for $\beta$ are known for the general convex case. We use the same $\beta$ for HB and ALR-MAG for direct comparison. For the step-size of HB we used $\gamma = 2(1-\beta)/L$ as recomended in Ghadimi et al. (2015). The results are presented in Figure 7. In this problem we observe that our step-size ($\text{MomPS}_{\max}$) is the fastest, having similar performace with ALR-MAG and GD Polyak. For the deterministic setting we have used a 12 core AMD Ryzen 5 5600H CPU to run the experiments.

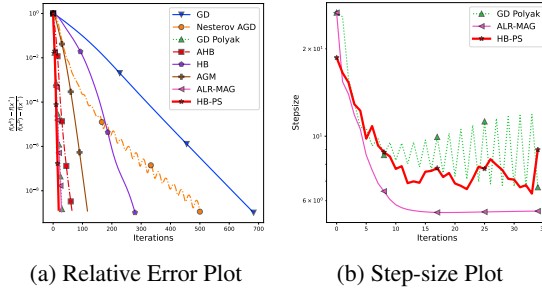

(a) Relative Error Plot          (b) Step-size Plot

Figure 7: Comparison of various deterministic algorithms for the logistic regression problem on synthetic data.

### F.2  STOCHASTIC HEAVY BALL WITH CONSTANT STEP-SIZE

As we saw in Corollary 3.5, when $\gamma_b \leq \frac{1}{2L_{\max}}$ then we have new theoretical guarantees for SHB with the constant step-size $\gamma = \frac{1-\beta}{2L_{\max}}$. The most recent analysis of constant SHB is from Liu et al. (2020) with step-size $\gamma = \frac{(1-\beta)^2}{L} \min\left\{\frac{1}{4-\beta+\beta^2}, \frac{1}{2\sqrt{2\beta+2\beta^2}}\right\}$. As mentioned in the main text our step-size is larger when $\beta \geq \sqrt{5} - 2 \approx 0.236$. In Figure 8 we provide a numerical comparison of these results for logistic regression with synthetic data. We observe that SHB with the Liu et al. (2020) step-size has faster convergence for the first iterations but it reaches a plateau much earlier. Moreover, note that as $\beta \to 1$ both step-sizes have similar performance.

### F.3  OTHER CHOICES

In this paper, we have provided convergence guarantees for SHB update rule $x^{t+1} = x^t - \gamma_t \nabla f_{S_t}(x^t) + \beta(x^t - x^{t-1})$ with the $\text{MomSPS}_{\max}$ step-size given by $\gamma_t = (1 - \beta) \min\left\{\frac{f_{S_t}(x^t) - \ell_{S_t}^*}{c\|\nabla f_{S_t}(x^t)\|^2}, \gamma_b\right\}$. However, as discussed in the main paper a more natural step-size would be to choose the well known $\text{SPS}_{\max}$ on the SHB rule. We call this new update rule $\text{SPS}_{\max}$ with naive momentum. Here we numerically compare these two updates. As we see in Figure 9, when $\beta$ is "small" and close to 0 then both $\text{SPS}_{\max}$ with naive momentum and $\text{MomSPS}_{\max}$ are close in performance. Of course, when $\beta = 0$ they are both equal to standard $\text{SPS}_{\max}$. However, as $\beta$ gets larger the performance of $\text{SPS}_{\max}$ with naive momentum gets worse and less stable and at some point it diverges. For $\beta \geq 0.7$ there is no $\text{SPS}_{\max}$ with naive momentum, since it cannot be numerically computed.

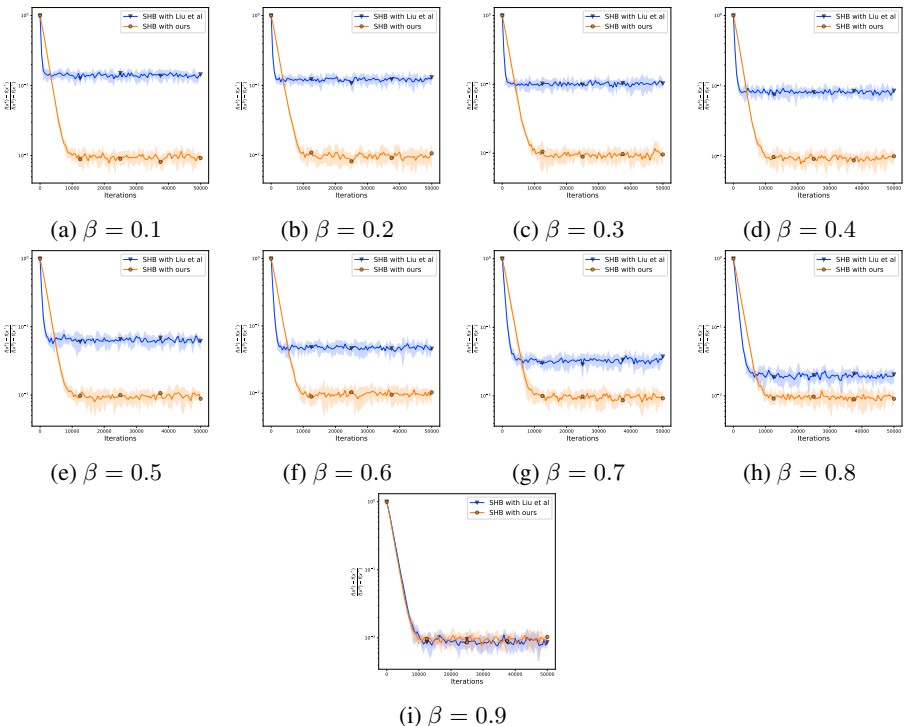

Figure 8: Comparison of SHB with constant step-size. (Liu et al., 2020) vs Corollary 3.5 for various momentum coefficients $\beta$ on logistic regression with synthetic data.

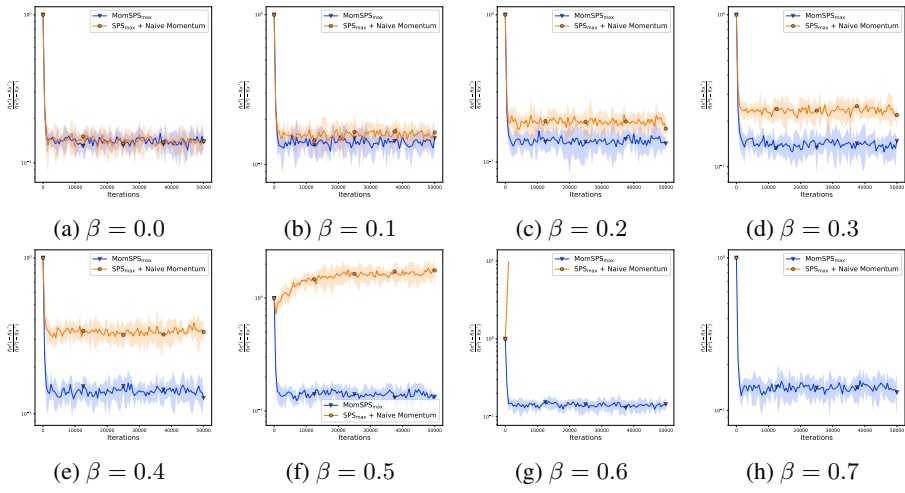

Figure 9: Comparison of MomSPS$_{\max}$ versus SPS$_{\max}$ with naive momentum for various momentum coefficients $\beta$ on logistic regression with synthetic data.

Furthermore, recall that our decreasing variants MomDecSPS and MomAdaSPS are respectively given by

$$\gamma_t = \min\left\{ \frac{(1-\beta)[f_{S_t}(x^t) - \ell^*_{S_t}]}{c\sqrt{t+1}\|\nabla f_{S_t}(x^t)\|^2}, \frac{\gamma_{t-1}\sqrt{t}}{\sqrt{t+1}} \right\}, \tag{MomDecSPS}$$

$$\gamma_t = \min\left\{ \frac{(1-\beta)[f_{S_t}(x^t) - \ell^*_{S_t}]}{c\|\nabla f_{S_t}(x^t)\|^2\sqrt{\sum_{s=0}^{t} f_{S_s}(x^s) - \ell^*_{S_s}}}, \gamma_{t-1} \right\}. \tag{MomAdaSPS}$$

The above step-sizes are theoretically inspired by IMA, as explained in Appendix C.3, but again a natural question is what happens if we take the "correcting factor" $1 - \beta$ outside the minimum. We call these step-sizes Alternative MomDecSPS and Alternative MomAdaSPS and are given by

$$\gamma_t = (1 - \beta) \min \left\{ \frac{f_{S_t}(x^t) - \ell^*_{S_t}}{c\sqrt{t+1}\|\nabla f_{S_t}(x^t)\|^2}, \frac{\gamma_{t-1}\sqrt{t}}{\sqrt{t+1}} \right\}, \qquad \text{(Alt MomDecSPS)}$$

$$\gamma_t = (1 - \beta) \min \left\{ \frac{f_{S_t}(x^t) - \ell^*_{S_t}}{c\|\nabla f_{S_t}(x^t)\|^2\sqrt{\sum_{s=0}^{t} f_{S_s}(x^s) - \ell^*_{S_s}}}, \gamma_{t-1} \right\}. \qquad \text{(Alt MomAdaSPS)}$$

In Figure 10 and Figure 11 we see that when $\beta = 0$ we have similiar performances, however when $\beta > 0$ then our proposed step-sizes are significantly better. A possible reason is that for both Alt MomDecSPS and Alt MomadaSPS it holds $\gamma_t \leq (1 - \beta)\gamma_{t-1}$ for all $t$. Thus inductively we get $\gamma_t \leq (1 - \beta)^t \gamma_0$. So when $\beta > 0$, $(1 - \beta)^t \to 0$ as $t \to \infty$ which means that the step-size is too small and it barely updates $x^t$.

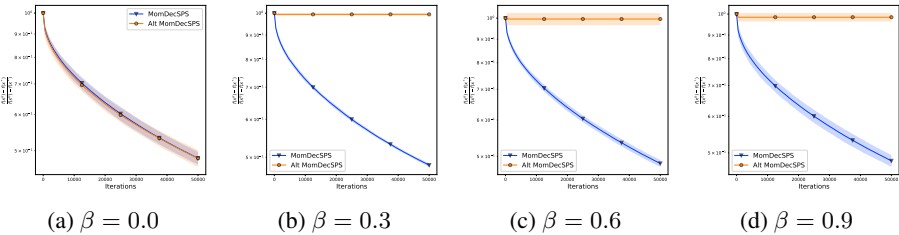

(a) $\beta = 0.0$      (b) $\beta = 0.3$      (c) $\beta = 0.6$      (d) $\beta = 0.9$

Figure 10: Comparison of MomDecSPS versus Alternative MomDecSPS for various momentum coefficients $\beta$ on logistic regression with synthetic data.

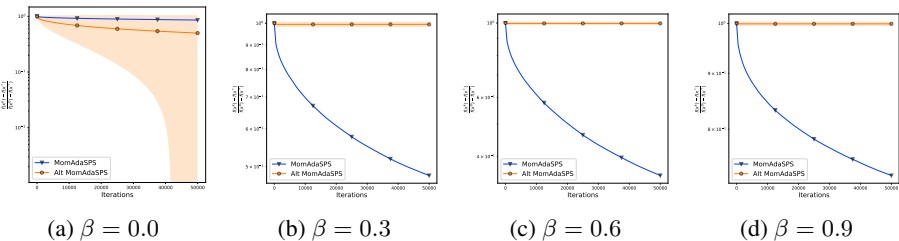

(a) $\beta = 0.0$      (b) $\beta = 0.3$      (c) $\beta = 0.6$      (d) $\beta = 0.9$

Figure 11: Comparison of MomAdaSPS versus Alternative MomAdaSPS for various momentum coefficients $\beta$ on logistic regression with synthetic data.

### F.4 IS $\gamma_b$ NEEDED?

Recall the definition of MomPS$_{\max}$, $\gamma_t = (1 - \beta) \min \left\{ \frac{f(x^t) - f(x^*)}{\|\nabla f(x^t)\|^2}, \gamma_b \right\}$. The more natural choice would be without the upper bound $\gamma_b$. Moreover, as in Appendix F.3, we can have the naive version of both of the above step-sizes, i.e. with no correcting factor $1 - \beta$. We compare the following step-sizes on both the least squares problem, in Figure 12, and logistic regression problem, in Figure 13, in the deterministic setting on synthetic data. The method is SHB with $\beta = 0.97$ for the least squares and $\beta = 0.3$ for the logistic regression.

$$\gamma_t = (1 - \beta)\frac{f(x^t) - f(x^*)}{\|\nabla f(x^t)\|^2} \qquad \text{(MomPS)}$$

$$\gamma_t = \frac{f(x^t) - f(x^*)}{\|\nabla f(x^t)\|^2} \qquad \text{(Alt MomPS)}$$

$$\gamma_t = (1 - \beta) \min \left\{ \frac{f(x^t) - f(x^*)}{\|\nabla f(x^t)\|^2}, \gamma_b \right\} \qquad \text{(MomPS}_{\max}\text{)}$$

$$\gamma_t = \min\left\{\frac{f(x^t) - f(x^*)}{\|\nabla f(x^t)\|^2}, \gamma_b\right\}. \qquad \text{(Alt MomPS}_{\max}\text{)}$$

We see that if $\gamma_b$ is chosen large enough, all these step-sizes have similar performance. However, we highlight the fact that convergence guarantees are known only for MomPS$_{\max}$.

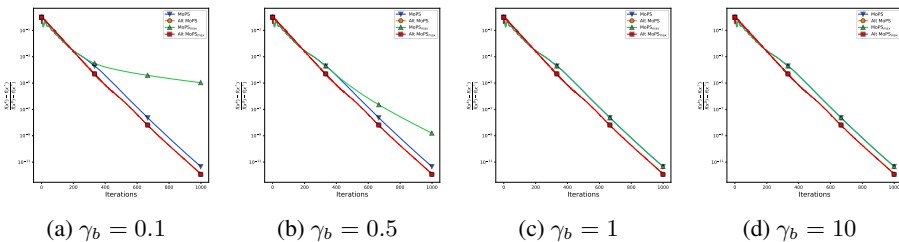

(a) $\gamma_b = 0.1$      (b) $\gamma_b = 0.5$      (c) $\gamma_b = 1$      (d) $\gamma_b = 10$

Figure 12: Comparison of various upper bounds for MomPS$_{\max}$ and other alternatives on least squares with synthetic data.

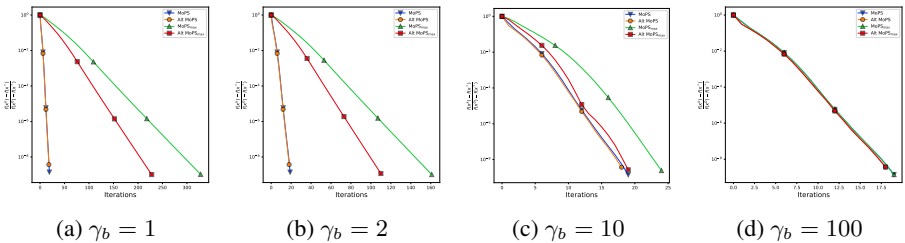

(a) $\gamma_b = 1$      (b) $\gamma_b = 2$      (c) $\gamma_b = 10$      (d) $\gamma_b = 100$

Figure 13: Comparison of various upper bounds for MomPS$_{\max}$ and other alternatives on logistic regression with synthetic data.

### F.5 PLOTS FOR $C_1$ AND $C_2$

Below it a plot of the constants $C_1$ and $C_2$ as functions of $\beta$ with $\gamma_b = 2$ and $\alpha = 1$ (recall $\gamma_b \geq \alpha$) from Theorem 3.2. Both functions are hyperbolas and the vertical line is the constant $\beta = \frac{\alpha}{2\gamma_b - \alpha}$. We see that in the interval $\beta \in \left[0, \frac{\alpha}{2\gamma_b - \alpha}\right)$ both functions are increasing.

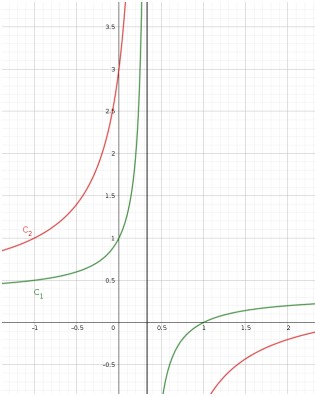

Figure 14: Plots for the constants $C_1$ and $C_2$ as functions of $\beta$. Here we have chosen $\gamma_b = 2$ and $\alpha = 1$.

## F.6 MORE CONVEX EXPERIMENTS AND PARAMETER SETTINGS

In this section, we list the parameters, architectures and hardware that we used for the deep learning experiments. The information is collected in Table 2. We also include some extra experiments in Figures 15 to 18.

| Hyper-parameter | Value |
| --- | --- |
| Architecture | Logistic Regression |
| GPUs | 1x NVIDIA GeForce RTX 3050 |
| Batch-size | See caption of each plot |
| Epochs | 100 |
| Trials | 5 |
| Weight Decay | 0.0 |

Table 2: Logistic regression experiment

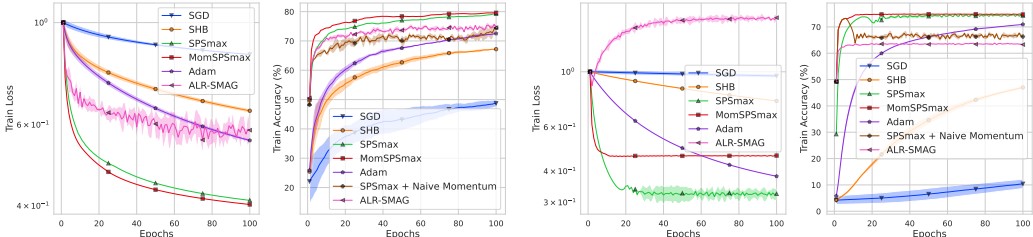

Figure 15: LibSVM dataset: vehicle, Batch size: 16    Figure 16: LibSVM dataset: letter, Batch size: 256

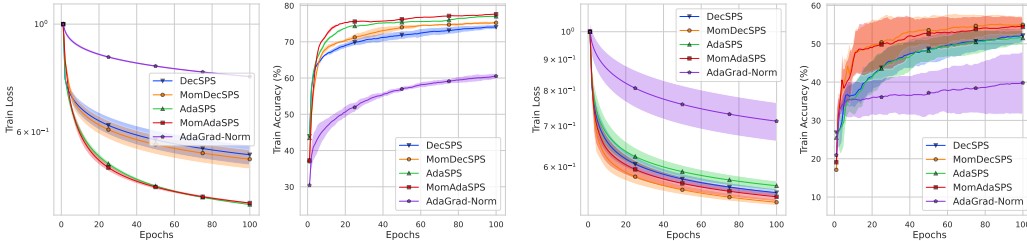

Figure 17: LibSVM dataset: vehicle, Batch size: 85    Figure 18: LibSVM dataset: glass, Batch size: 32

## F.7 MORE DEEP LEARNING EXPERIMENTS AND PARAMETER SETTINGS

In this section, we list the parameters, architectures and hardware that we used for the deep learning experiments. The information is collected in Table 3. We also include some extra experiments in Figures 19 and 20. For ALR-SMAG we use $c = 0.1$ in the DL experiments.

| Hyper-parameter | Value |
| --- | --- |
| Architecture | ResNet 10/34 (He et al., 2016) |
| GPUs | 1x Nvidia RTX 6000 Ada Generation |
| Batch-size | 256 |
| Epochs | 100 |
| Trials | 5 |
| Weight Decay | 0.0 |

Table 3: CIFAR10 experiment

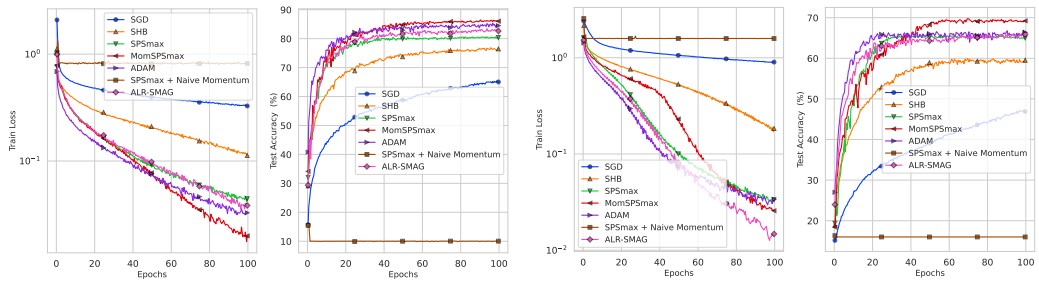

Figure 19: Resnet 18 on CIFAR 10        Figure 20: Resnet 18 on CIFAR 100

