# OpenReview forum: "Stochastic Polyak Step-sizes and Momentum: Convergence Guarantees and Practical Performance"
_ICLR.cc/2025/Conference — ICLR 2025 Poster_

### Official Review · Reviewer_jVKN · 2024-10-23

**Soundness:** 3
**Presentation:** 3
**Contribution:** 3
**Rating:** 6
**Confidence:** 3

**Summary:**

This paper studies stochastic Polyak step-size for SHB. Three step size selections are proposed and associated convergence guarantees are developed. Specifically, using the MomSPSmax stepsize, the SHB is proved to converge to a neighborhood of the solution, if the objective function is convex and smooth. Moreover, another two stepsizes, I.e., MomDecSPS and MomAdaSPS, are proposed for non-interpolated SHB, which guarantee that SHB iterates converging to the exact solution. Finally, numerical experiments are presented to show the effectiveness of the proposed methods.

**Strengths:**

1.The writing and exposition are clear.

2.The proposed methods are practical to implement and have theoretical improvements compared with existing works.

**Weaknesses:**

Most of techniques are from existing literature. The authors did not explain the difficulty of applying the Polyak step-size to SHB compared with the SGD. This makes the technique used in this paper look like a direct generalization of Loizou et al. (2021)

**Questions:**

None.

---

> ### Author Response · Authors · 2024-11-21
> **Authors' response to Reviewer jVKN**
>
> We would like to thank the reviewer for their time and positive evaluation. Below, we address the concern raised by the reviewer.
>
> Stating that our approach or techniques are the same as [Loizou, 2021] is comparable to claiming that all analyses of SGD with momentum are equivalent to those of vanilla SGD. Our work builds on previous techniques but introduces significant differences that result in new theoretical contributions and practical implications.
>
> **On proof techniques:**
>
> Compared to the analysis of SGD with SPS from [Loizou, 2021], the SHB update rule requires taking into consideration extra terms related to the previous iterate $x^{t-1}$. This needs new algebraic techniques in order to use convexity, and after that, one needs to deal with the two quantities $f(x^t)$ and $f(x^{t-1})$ at the same time. We deal with this using the IMA framework, similar to Sebbouh [2021]. However, we highlight that we finish our proofs with a different approach. More specifically, in the proof of Theorem G.1 in Sebbouh [2021] the authors choose the momentum coefficients $\lambda_t$ in such a way that the main sum telescopes and the final result only has the expectation of the last iterate. In our proof, we make no such simplifications, and instead, we use Jensen’s inequality to finish. This approach allows us to have more freedom in the selection of $\lambda_t$. This is more prominent in the proofs of the decreasing variants.
>
> Compared to the classical analysis of constant step-size SHB, the use of SPS requires an adaptive step-size that uses the loss and stochastic gradient estimates at an iterate, resulting in correlations of the step-size with the gradient, i.e. we might have $\mathbb{E}[\gamma_t\nabla f_i(x^t)]\neq\gamma_t\mathbb{E}[\nabla f_i(x^t)]$. One of the technical challenges in the proofs is to carefully analyze the SHB iterates, taking these correlations into account. Moreover, since we try to be adaptive to the Lipschitz constant, we can not use any standard descent lemmas (implied by the smoothness and the SHB update) that are used in the classical analysis of SGD and constant step-size SHB. This makes the convex proof more challenging than the standard analysis of SHB.
>
>
> Building on the analysis of previous works should not be considered a drawback or a reason to question the novelty of a paper. In the end, a paper should be valuable if it answers open questions in the literature, and we believe that our work does that.
> **If you agree that we managed to address all issues, please consider raising your mark to support our work. If you believe this is not the case, please let us know so that we have a chance to respond.**

---

> ### Comment · Reviewer_jVKN · 2024-11-24
>
> The authors' rebuttal basically addresses my concern. I increase the contribution score to 3 and maintain the overall score.

---

### Official Review · Reviewer_aQjo · 2024-10-27

**Soundness:** 3
**Presentation:** 3
**Contribution:** 3
**Rating:** 8
**Confidence:** 3

**Summary:**

The authors introduce new step-size schedules which are inspired by stochastic Polyak step-size (SPS) for the Stochastic Heavy Ball (SHB) momentum optimization method. The authors present convergence rates for each of their step-size selections showing that SHB can converge to the true solution at a fast rate when interpolation is satisfied and converge to the minimizer adaptively when interpolation is not satisfied.

**Strengths:**

The paper effectively integrates SPS with SHB, an approach previously applied to SGD in the existing literature. The proposed methods are validated both theoretically and empirically by demonstrating their convergence rates and conducting relevant experiments.

**Weaknesses:**

Gaining understanding of SHB within SPS framework and establishing a convergence rate is a valuable contribution. However, identifying the specific settings or problem types where the proposed method is most effective would provide a more comprehensive and practical understanding of its applicability. Based on my current understanding of the methods, from a theoretical perspective, there does not appear to be a clear advantage for incorporating momentum within the SPS framework.

**Questions:**

In the convex setting, the paper indicate that no momentum produce the best convergence rate (line 354), the convergence rate in Theorem 3.5 and Theorem 3.6 deteriorate when $\beta$ increases. Hence, within the SPS framework, which specific settings or problem types momentum should be employed instead of relying solely on SGD?

The assumption of bounded iterates in Section 3.2 constitutes a strong condition, which may not hold even for quadratics without additional constraints. In the absence of this assumption, is it possible to derive meaningful theoretical results for the MomDecSPS and MomAdaSPS methods?

---

> ### Author Response · Authors · 2024-11-21
> **Authors' response to Reviewer aQjo**
>
> We thank the reviewer for taking the time to review our work. We appreciate that the reviewer finds our convergence rates “a valuable contribution”. Below, we address all the concerns and questions raised.
>
> **On using momentum:**
>
> The main advantage of momentum lies in increasing the generalization performance of an algorithm in applications. In all of the experiments of the paper, we demonstrate that our MomSPS variant provides significant benefits compared to the no-momentum variants, which is not achieved by simply applying standard momentum to SPS-type methods. We agree with the reviewer that theoretically, our results do not show an advantage of using momentum on top of Polyak-type stepsizes. However, to the best of our knowledge, this is the case with all existing analyses of momentum on top of vanilla SGD. The benefit of the momentum in all existing works is practical and typically leads to better generalization. This should not undermine the main contribution of our work which is the understanding that the naive use of momentum on top of Polyak-type stepsizes might fail to converge and the proposal of the IMA viewpoint that allows us to resolve the issue.
>
> **On bounded iterates assumption and projection:**
>
> Indeed our two main theorems on the convergence of MomDecSPS/MomAdaSPS require the bounded iterates assumption, however, the results are meaningful. This is exactly the assumption used in the original papers by [Orvieto, 2022] and [Jiang, 2023] that proposed the analyses of the no-momentum variants (DecSPS/AdaSPS). In these papers, it is possible to remove the bounded iterates assumption by assuming strong convexity of individual f_i. This is a strong alternative assumption that holds for a much more restrictive setting than the general convex regime, which is the main focus of our work. We speculate that with this extra assumption, a similar result might be possible for the MomDecSPS/MomAdaSPS too.
>
> Another alternative to removing this assumption is by projection: Our results can be extended to the constrained regime by adding a projection step in the update rule of IMA onto a convex and bounded domain D. That is $z^{t+1}=proj_D(z^t-\eta_t\nabla f_{S_t}(x^t))$ with $x^0=z^0\in D$. Note that the projection step is only needed for the update of $z^{t+1}$ because $x^{t+1}$ is already a convex combination of elements in the convex set D. Our current proofs will then go through using the non-expansiveness of the projection operator.
>
> **If you agree that we managed to address all issues, please consider raising your mark. If you believe this is not the case, please let us know so that we have a chance to respond.**

---

> > ### Comment · Reviewer_aQjo · 2024-11-23
> >
> > Thank you for your response. The authors addressed my questions and concerns. It would be great if the authors can add the proof with projection and some related empirical experiments if possible in the appendix and mention that in the main paper. Currently, I am thinking of adding 1 to the score from 6 to 7 but there is no option for that. Given that there are other reviewers also have similar concern about the bounded iterates assumption, if the author manage to update a revised version of the paper with the complete proof using projection, I can give the score 8.

---

> > > ### Author Response · Authors · 2024-11-26
> > >
> > > We are glad that we covered all your questions and concerns.
> > >
> > > We have updated our draft to include the full description of the Projected IMA and the corresponding proofs (requested complete proof using projection) in Appendix E, named “Beyond the bounded iterates assumption”. Please let us know if the final concerns related to the bounded iterates assumption have been addressed.

---

> > > > ### Author Response · Authors · 2024-11-28
> > > >
> > > > Dear Reviewer **aQjo**,
> > > >
> > > > The deadline for updating the pdf is in a few hours.
> > > > As we mentioned in our previous message, we have updated our draft to include the full description of the Projected IMA and the corresponding proofs (requested complete proof using projection) in Appendix E, named “Beyond the bounded iterates assumption”.
> > > >
> > > > Please let us know if the final concerns related to the bounded iterates assumption have been addressed.

---

> > > > > ### Comment · Reviewer_aQjo · 2024-11-28
> > > > >
> > > > > Thank you for your response and updated draft that include the projected proofs. I increased the score to 8. In later revision, I would also like to see how projected IMA performs compare to other projected methods in empirical evaluation.

---

### Official Review · Reviewer_QP47 · 2024-11-02

**Soundness:** 3
**Presentation:** 3
**Contribution:** 3
**Rating:** 6
**Confidence:** 3

**Summary:**

This paper extends the recently developed Stochastic Polyak Stepsize (SPS) to incorporate the Pokyak momentum. In particular, the paper analyzes three SPS variants, vanilla SPS, DesSPS, and AdaSPS, and gets convergence guarantee for their momentum variants: MomSPS, MomDesSPS, and MomAdaSPS. Extensive numerical experiments demonstrate the practical performance of the proposed methods.

**Strengths:**

The paper is well-written and solid, with details and intuitions clearly explained to the readers. Moreover, extensive experiments validate the practical performance of the proposed stepsizes.

**Weaknesses:**

Although the paper is a solid work, my foremost concern is its limited technical novelty.

1. Limited technical contributions

   SPS [1], DesSPS [2], and AdaSPS [3] considered in this paper are already present in the literature, and the contribution of the paper only lies in showing the feasibility of incorporating momentum into these existing stepsizes.

2. Strong assumptions

   The analysis of MomSPS imposes a constraint on $\beta$, and the analysis of two other stepsizes needs bounded iterates. Although the authors claim these are standard assumptions, they do not usually appear in the literature for SHB methods.

Overall, despite the limited theoretical novelty due to the strong assumptions, I believe this paper is a solid contribution to the SPS literature and I recommend weak acceptance.

**Questions:**

**Questions**

1. The comparison after Corollary 3.4 looks unclear to me. In particular, the results in [6] can achieve $O(1/\sqrt{K})$ convergence in the presence of noise by taking $\alpha =O( 1/\sqrt{K})$ (Theorem 1, [6]) , while SPS cannot achieve exact convergence. Could you elaborate more on this comparison?
2. Momentum in stochastic optimization is often shown to achieve certain variance reduction effects [4, 5]. Do you think this can help improve dependence on $\sigma^2$ in the convergence analysis of MomSPS?

**Minor issues**

1. Line 55

   has efficiently analyzed => has been efficiently analyzed.

2. Line 400, 410

   Please be consistent with step size and step-size.

3. Line 1098

   Definition C.2. Smoothness requires a two-sided bound $|f(x) - f(y) - \langle \nabla f(y), x- y\rangle | \leq \frac{L}{2}\\|x - y\\|^2 $. Concave functions without Lipschitz continuous gradient can satisfy the given one-sided bound.

**References**

[1] Nicolas Loizou, Sharan Vaswani, Issam Hadj Laradji, and Simon Lacoste-Julien. Stochastic polyak step-size for sgd: an adaptive learning rate for fast convergence. In *International Conference on Artificial Intelligence and Statistics*, pages 13061314. PMLR, 2021.

[2] Antonio Orvieto, Simon Lacoste-Julien, and Nicolas Loizou. Dynamics of sgd with stochastic polyak stepsizes: truly adaptive variants and convergence to exact solution. *Advances in Neural Information Pro- cessing Systems*, 35:2694326954, 2022.

[3] Xiaowen Jiang and Sebastian U Stich. Adaptive sgd with polyak stepsize and line-search: robust convergence and variance reduction. *Advances in Neural Information Processing Systems*, 36, 2024.

[4] Cutkosky, Ashok, and Francesco Orabona. Momentum-based variance reduction in non-convex sgd. *Advances in neural information processing systems* 32, 2019.

[5] Gao, Yuan, Anton Rodomanov, and Sebastian U. Stich. Non-Convex Stochastic Composite Optimization with Polyak Momentum. In *Forty-First International Conference on Machine Learning*, 2024.

[6] Liu, Yanli, Yuan Gao, and Wotao Yin. An improved analysis of stochastic gradient descent with momentum. *Advances in Neural Information Processing Systems* 33, 2020.

---

> ### Author Response · Authors · 2024-11-21
> **Authors' response to Reviewer QP47**
>
> We thank the reviewer for taking the time to review our work. We appreciate that the reviewer finds our paper “solid work”. Below, we address all the concerns and questions raised.
>
> **Limited technical contributions:**
>
> Building on the analysis of previous works should not be considered a drawback or a reason to question the novelty of a paper. In the end, a paper should be valuable if it answers open questions in the literature and we believe that our work does that.
> That being said, in this work, we do not just blindly take the previous stepsizes and we prove that they also work in the SHB setting. This is the naive approach we discuss in Section 2.2 and we demonstrate that it might lead to divergence even in simple scenarios. We use the IMA framework to derive our stepsizes and to make sure they depend on the momentum coefficient β. Furthermore, the proofs are not straightforward adaptations from the SGD case. Compared to the analysis of SGD with SPS, the SHB update rule requires taking into consideration extra terms related to the previous iterate $x^{t-1}$. This needs new algebraic tricks in order to use convexity, and after that, one needs to deal with the two quantities $f(x^t)$ and $f(x^{t-1})$ at the same time.
>
> **Strong assumptions:**
>
> Indeed our two main theorems on the convergence of MomDecSPS/MomAdaSPS require the bounded iterates assumption. This is exactly the assumption used in the original papers by [Orvieto, 2022] and [Jiang, 2023] that proposed the analyses of the no-momentum variants (DecSPS/AdaSPS). In these papers, it is possible to remove the bounded iterates assumption by assuming strong convexity of individual f_i. This is a strong alternative assumption that holds for a much more restrictive setting than the general convex regime, which is the main focus of our work. We speculate that with this extra assumption, a similar result might be possible for the MomDecSPS/MomAdaSPS too.
>
> Another alternative to removing the bounded iterates assumption is via projection: Our results can be extended to the constrained regime by adding a projection step in the update rule of IMA onto a convex and bounded domain D. That is $z^{t+1}=proj_D(z^t-\eta_t\nabla f_{S_t}(x^t))$ with $x^0=z^0\in D$. Note that the projection step is only needed for the update of $z^{t+1}$ because $x^{t+1}$ is already a convex combination of elements in the convex set D. Our current proofs will then go through using the non-expansiveness of the projection operator.
>
> **[Q1]:**
> Let’s focus firstly on the no momentum case (ie β=0) where SHB reduces to SGD. SGD with constant stepsizes converges up to a neighborhood while SGD with decreasing stepsizes converges to exact solution with rate $O(1/\sqrt{K})$. A similar thing is true for SPS. SGD with SPSmax converges up to a neighborhood while SGD with DecSPS converges to an exact solution with rate $O(1/\sqrt{K})$. All of these can be transferred to SHB. As you mentioned SHB with decreasing stepsizes ( $a=O(1/\sqrt{K})$) converges to exact solution with rate  $a=O(1/\sqrt{K})$. In Thm 3.5, we show that SHB with MomDecSPS (and MomAdaSPS) converges to the exact solution again with rate  $a=O(1/\sqrt{K})$.
>
> **[Q2]:**
> This is an intriguing question. We do not know the answer to this, however, it seems like an interesting future research direction.
>
> **Minor issues:** Thanks for catching them. All of them will be fixed.
>
> We are glad that the reviewer recognizes our contribution to the SPS literature and would like to reiterate that our work addresses broader regimes with no stronger assumptions than prior studies, making it a significant theoretical advancement. We hope that with our response, we will clarify the theoretical novelty of our results further and explain the assumptions used (no stronger than previous works on adaptive methods).
> **If you agree that we managed to address all your questions, please consider raising your mark to support our work. If you believe this is not the case, please let us know so that we have a chance to respond.**

---

> > ### Comment · Reviewer_QP47 · 2024-11-22
> >
> > Thank you for the response. The authors addressed several of my concerns. I feel the score of the paper is somewhere between 6 and 8, but given that there is no option of 7, I will maintain the current score and increase my score of contribution from 2 (fair) to 3 (good).

---

### Official Review · Reviewer_SmTn · 2024-11-04

**Soundness:** 3
**Presentation:** 2
**Contribution:** 3
**Rating:** 8
**Confidence:** 2

**Summary:**

This paper introduces Polyak step size to Stochastic Heavy Ball method. To do this, authors consider iterate moving-avergage viewpoint of stochastic heavy ball method. With slight modification of the step size adjustment technique, authors propose three new step size selection: one is based on regular Polyak step size for SGD, and the rest - on decreasing versions of Polyak step size for SGD. The resulting algorithms do not need knowledge about any of problem parameters and converge either to exact solution in non-interpolated regime or to the area of solution in interpolated regime. Convergence rates match the existing state-of-the-art results.

**Strengths:**

Authors propose a connection between both Polyak step size and adaptive Polyak step size with stochastic heavy ball method, which seems like an important result for optimization community. The proposed technique does not need any knowledge about problem parameters, which makes methods easier to implement. Theoretical convergence results show that proposed methods either outperform or perform on the same level with existing state-of-the-art algorithms while sometimes mitigating some of the issues of these algorithms. Finally, experimental results show either similar or better performance on both synthetic and real problems in comparison with existing analogs.

**Weaknesses:**

The Section 3 is written in rather overwhelming way. After each theorem goes a paragraph, mentioning some special cases and how the provided result matches the results from other existing works, which is rather hard to comprehend for a new reader. You could change these paragraphs to small tables with small supporting captions. This should take about the same size as the paragraph in text, although it would be much more evident. Additionally, after every theorem follows a lot of corollaries, that again consider some special cases. It is easy to get lost in so many special cases. I think, some of these corollaries can also be moved to appendix, leaving some mention about them in the main part. Since the main result is introduction of Polyak step size and adaptive Polyak step size to SHB, it should be the central point of main part of the paper.
Overall, despite the seemingly good results, considering everything above and some suggestions in Question part, I think the main text needs polishing and not ready yet for publication.

**Questions:**

## Questions
1. Am I correct, that by "interpolated regime" you mean that noise variance is zero and we have determenistic case?
1. Figures 8 and 9. In first figure you have orange color for your method, in the second - blue. This is confusing, please, fix the order of the plots in all the figures for consistency. Also in Figure 8 you do not use the name of your particular algorithm. Please also check, that these issues are fixed for all other figures.
1. Lines 382-383. What do you mean by "saddle connection"?

## Suggestions
1. Please, add explicit assumptions in the text about interpolation and smoothness etc., that you use in the theorems in some section like "Preliminaries". Sometimes it is hard to follow, whether you mean interpolation or bounded variance and it is hard to find them in the text.
1. Please, increase font size in the figures.

---

> ### Author Response · Authors · 2024-11-21
> **Authors' response to Reviewer SmTn**
>
> We thank the reviewer for taking the time to review our work. Below, we address all the questions raised.
>
> We respectfully disagree with the reviewer’s statement that the paper is “not ready yet for publication” due to its presentation. This is a theory/optimization paper, and our structure aligns with the standard conventions for papers in this field, emphasizing precise and rigorous statements in the main body. Moreover, two other reviewers (QP47 and jVKN) have explicitly commented on the clarity of the exposition and noted that the paper is well-written. Furthermore, the reviewer suggested adding tables; however, we already include a table (Table 1) that provides informal summaries of our theorems alongside comparisons to related works. Additionally, regarding the suggestion for a “Preliminaries” section, we do have such a section in Appendix C, titled “Technical Preliminaries,” where all details are presented (due to space limitation, such details are typically included in the appendix)
>
> About the reviewer’s Questions:
>
> **[Q1]:** You are correct that interpolation means that the noise variance ($\sigma^2$) is zero. However, this does not necessarily imply that we are in the deterministic case. The deterministic case is only a special case of a setting where the variance is zero. Besides that, there are plenty of stochastic interpolated examples. As mentioned in the paper, non-parametric regression (Liang and Rakhlin, 2020) and over-parameterized deep neural networks (Zhang et al., 2021; Ma et al., 2018) satisfy the interpolation condition.
>
> **[Q2]:** Thanks for catching these. These can be easily fixed.
>
> **[Q3]:** The word saddle can be removed.
>
> As we highlighted above, the current structure of our paper and its presentation effectively communicates the content. We believe that the presentation is a personal style of the authors of each paper, and one should not force their preferences on them. Providing feedback is, of course, welcome for improving details on the exposition, and we appreciate the comments. However, claiming that a theoretically focused paper should not provide details of the main theoretical statements as corollaries and remarks (a common practice in the majority of optimization/theory-oriented papers) after the main Theorems in the main paper is not constructive feedback.
>
> We agree with the reviewer that the main result is introducing the adaptive Polyak step size to SHB, and we devote section 2 (more than 3 pages of the main paper) to exactly the introduction of the main ideas and the connection with what exists. However, just the introduction of the new ideas is not enough; one needs a detailed exposition of the convergence guarantees of the method in different variants and settings. This is when the main theoretical results thrive compared to what is known in the literature. At first glance, Section 3 might look overwhelming for the reviewer, but it is divided into sections with the main theorems highlighting the importance of each result and the discussion after explaining the benefits and limitations of the proposed analysis. This is done by design so that experts in the area can read and understand exactly the importance of each statement.
>
> The only concern raised as a weakness of our work is the presentation of Section 3. As you argue, this is a personal taste of the authors of each paper and should not be considered a reason to suggest rejection. Please see the other reviewers that highlight the quality of this section and how well-written it is.
> **If you agree that we managed to address your concers, please consider raising your mark. If you believe this is not the case, please let us know so that we have a chance to respond.**

---

> > ### Comment · Reviewer_SmTn · 2024-11-25
> >
> > I thank the authors for answering my questions. However, let me kindly disagree with them on some points, considering the text structure. When I read a paper, I want to understand the main results of the paper and whether they are relevant to me as fast as possible. This means skimming through assumptions, theorem formulations, corollaries, and numerical results. In this paper it becomes tricky. For example, in Section 3.1 the authors provide the main assumption about finite optimal objective difference in the main text and do not formulate it as separate "Assumption N", or in Corollary 3.4 the authors do not mention, that they consider constant step size here, which may be confusing if the reader did not read preceding paragraph. It should be noted, that paper [1], which the authors cite and that has a similar structure to this paper, lacks such problems. I understand, that most likely this was done because of the space economy, but I think this does not do well when we talk about readability. That is why I asked about the Preliminaries section in the main part. If the authors chose to move this section to the Appendix, they could also move the assumption about finite optimal objective difference and just \ref it in the theorem formulations. This should not take that much space but will improve readability.
> >
> > I agree that sometimes the authors' presentation of the results is a personal style, but the authors still need to think about the reader. As for me, it was not very easy to comprehend Section 3, because I'm not very familiar with works, considering finite optimal objective difference (that's why I have confidence 2). That is why I think the points I mention here are important to help readers from adjacent areas understand the results of the paper faster.
> >
> > Speaking of my other remarks, I agree, that they were not that important.
> >
> > [1] Loizou, Nicolas, et al. "Stochastic polyak step-size for sgd: An adaptive learning rate for fast convergence." International Conference on Artificial Intelligence and Statistics. PMLR, 2021.

---

> > > ### Author Response · Authors · 2024-11-26
> > >
> > > We thank the reviewer for the follow-up clarifications. Now, we understand better what they refer to in terms of better readability and easier access to the main assumptions of our results and the main outcome of each corollary. To accommodate their suggestion, we now include the finite optimal objective difference in an assumption (see Assumption 3.1) in our paper and add titles for all corollaries in our main paper (see updated pdf).
> > >
> > > As we mentioned in our previous message, the only concern raised as a weakness of our work is the presentation of Section 3. We still stand by our claim that this is a personal taste of the authors of each paper and should not be considered a reason to suggest rejection. However, we appreciate the more precise feedback given to us and make the necessary changes.  We highlight one more time that the other reviewers mentioned that this section is very well written and has a clear presentation regarding the convergence guarantees of our methods.
> > >
> > > **If you agree that we managed to address your concerns, please consider raising your mark. If you believe this is not the case, please let us know so that we have a chance to respond.**

---

> ### Comment · Reviewer_SmTn · 2024-11-26
>
> The authors adressed all of my concerns. I increase my rating to 8, and presentation score - to 2.

---

### Author Response · Authors · 2024-11-21
**General response to all reviewers**

We thank the reviewers for their feedback and time.

In particular, we appreciate that the reviewers acknowledged the following strengths of our work:
* Reviewer **SmTn** acknowledges that our work is an important result for the optimization community and recognizes that our stepsizes do not need any knowledge about problem parameters, which makes methods easier to implement.
* Reviewer **QP47** finds our well-written, with details and intuitions clearly explained to the readers as well as acknowledges that it is solid work.
* Reviewer **aQjo** appreciates that our proposed stepsizes are validated both theoretically and empirically and finds our convergence rates a valuable contribution.
* Reviewer **jVKN** highlights that our writing and exposition are clear and that our proposed stepsizes have theoretical improvements compared with existing works.

With our rebuttal, we address all raised issues. **Here we highlight again that with our work:**

* We explain why naively using SPSmax as a step-size $\gamma_t$ in the update rule of the Stochastic Heavy Ball method (SHB) **leads to divergence** even in simple problems (see also Figure 1).
* To resolve this issue, we provide an alternative way of selecting Polyak-type step-sizes for SHB **via the Iterate Moving Average viewpoint, and we propose three novel adaptive step-size selections** (MomSPSmax, MomDecSPS, MomAdaSPS).
* MomSPSmax is an adaptive step-size selection that depends on the momentum parameter $\beta$. Most importantly, **SHB with MomSPSmax is the first adaptive SHB with convergence guarantees for the fully stochastic setting (no interpolation assumed).**
* Our decreasing variants, MomDecSPS and MomAdaSPS, are the **first adaptive step-sizes for SHB with convergence guarantees to the exact solution (in the non-interpolated regime)**. Furthermore, via MomAdaSPS we provide the first robust convergence of adaptive SHB that guarantees convergence to the exact solution and automatically adapts to whether our problem is interpolated or not.
* All the provided convergence guarantees are **tight** in the following sense: If $\beta=0$ (no momentum) in the update rule of SHB, our theorems recover as a special case the best-known convergence rates of Polyak-type step-size for SGD.
* Finally, we have extensive numerical evaluations both in the convex and non-convex regimes.

**We hope that you will engage with us in a back-and-forth discussion and we will be most happy to answer any remaining questions.**

---

### Meta-Review · Area_Chair_2QWw · 2024-12-18

**Metareview:**

Summary:
This paper extends Stochastic Polyak Stepsize (SPS) methods to incorporate momentum through the Stochastic Heavy Ball (SHB) method. The authors propose three novel adaptive stepsize selections (MomSPSmax, MomDecSPS, MomAdaSPS) and provide theoretical convergence guarantees. Most notably, MomSPSmax is the first adaptive SHB with convergence guarantees for the fully stochastic setting, while the decreasing variants are the first adaptive stepsizes for SHB that guarantee convergence to the exact solution in the non-interpolated regime.

Main Strengths:
The work provides a significant theoretical contribution by developing convergence guarantees that are tight and recover best-known rates for SGD as special cases. The proposed methods are practical to implement as they require no knowledge of problem parameters. The paper is well-written with clear exposition of technical details and intuition, supported by extensive numerical evaluations in both convex and non-convex settings.

Main Weaknesses:
Initial concerns were raised about the technical novelty given that the work builds on existing SPS methods. There were also questions about the strength of assumptions required, particularly the bounded iterates assumption. Some reviewers suggested improvements to the presentation of theoretical results to enhance readability.

**Additional Comments On Reviewer Discussion:**

Outcomes from Author-Reviewer Discussion:
The authors effectively addressed concerns by:

- Clarifying that the work is not a straightforward extension of existing methods, highlighting new technical challenges in analyzing SHB with adaptive stepsizes
- Providing alternative approaches to remove the bounded iterates assumption through projection methods, with complete proofs added to the appendix
- Making presentation improvements by adding clearer assumptions and theorem statements

Reviewer Agreement:
All reviewers acknowledged the paper's solid theoretical foundation and practical utility. While some initially questioned the technical novelty, the author responses satisfied these concerns, leading to improved evaluations of the paper's contribution. The final consensus supports acceptance, with ratings ranging from 6 to 8.

Suggestions to Improve:
The manuscript would benefit from:

- Including empirical evaluations of projected variants
- Further discussion of specific scenarios where momentum provides advantages
- Clearer organization of theoretical results and assumptions in the main text

The paper represents a valuable contribution to the optimization literature by developing the first adaptive momentum methods with convergence guarantees in important settings. The theoretical analysis is rigorous, and the practical utility is well-demonstrated. The authors have been responsive to reviewer feedback and made appropriate improvements.

---

### Decision · Program_Chairs · 2025-01-22

Accept (Poster)